# MLOT: Generalizing the Bipartite Structure to a Multi-Layered Framework for Optimal Transport

## Abstract

Although optimal transport (OT) has achieved significant success and widespread application in various fields, its structure remains relatively simple, relying on bipartite graphs with only two layers of nodes for transportation. In this paper, we propose a multi-layer optimal transport (MLOT) method that extends the original two-layer structure to handle transportation problems across multiple hierarchical levels, making it more adaptable to the complex structures found in deep learning tasks. In this framework, the source distribution flows through intermediate layers before reaching the target distribution, where estimating the intermediate distributions becomes crucial for solving the MLOT. Under entropic regularization, we further propose the MLOT-Sinkhorn algorithm to solve the multi-layer OT problem, where intermediate distributions can be estimated through the transportation calculations between adjacent layers. This algorithm can be accelerated using GPUs and significantly outperforms general solvers such as Gurobi. We also present theoretical results for the entropic MLOT, demonstrating its efficiency advantages and convergence properties. Furthermore, we find that our MLOT is well-suited for machine learning tasks based on data augmentation. As a result, we apply the MLOT-Sinkhorn algorithm to tasks such as text-image retrieval and visual graph matching. Experimental results show that reformulating these problems within the MLOT framework leads to significant improvements in performance.

## 1 Introduction

Optimal Transport (OT) [28] has been an increasingly important mathematical tool for solving various machine learning problems, with success in a wide range of applications, ranging from domain adaptation [36], learning generative models [3], network designing [43], self-supervised contrastive learning [6], to long-tail recognition [27] etc. It allows for the comparison of probability distributions, combining the underlying geometric structure of the sample space.

Based on entropic OT, the Sinkhorn algorithm, due to its GPU-friendly nature, allows for forward computation and backpropagation in neural networks, and has thus been widely used in deep learning tasks such as deep clustering, graph matching, and more. However, the vanilla OT behind the Sinkhorn algorithm typically relies on the assumption of a simple bipartite graph structure with fixed, known marginal distributions. This is problematic for tasks like zero-shot retrieval, where the distribution of the retrieved items (the target) is unknown. Standard approaches force a uniform prior on the target, which lacks physical significance and limits performance. By transitioning from a two-layer network to a multi-layered one, as shown in Fig. 1, MLOT allows us to treat these unknown distributions as latent intermediate layers that are adaptively computed, rather than fixed priors. Thereby broadening the OT theory and its potential applications for deep learning.

Thus, in this paper, we first propose a new variant of optimal transport called multi-layered optimal transport (MLOT) that extends the original two-layered transportation structure to the multi-layered case. As shown in Fig. 1, we assume the known source and target distributions in the source and target layers, along with the known cost matrices between layers. Our objective is to determine intermediate distributions and the transportation plan (i.e., coupling) between layers. Similar to vanilla OT, this problem fundamentally boils down to linear programming [9] and one can employ the (inefficient)

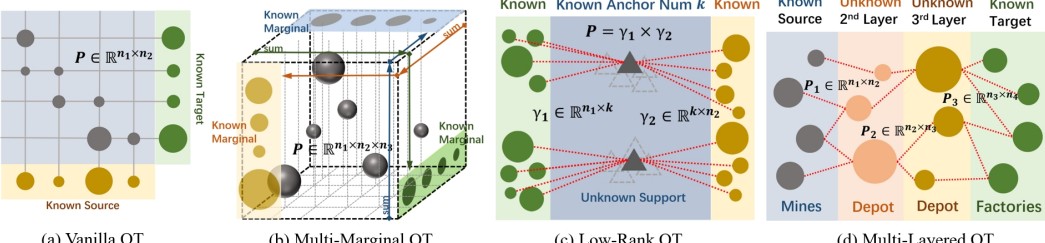

Figure 1: Comparisons among Vanilla OT, MMOT, LOT and MLOT(ours). Compared to the first two, the key focus of MLOT lies in the estimation of the unknown intermediate distributions. Relative to LOT, beyond the difference of employing a fixed support, they have more essential distinction in physical meaning: MLOT optimizes on real flow transition $\mathbf{P}_i$, while LOT optimizes probability transition $\boldsymbol{\gamma}_i$, which means the real flow from $i \to j$ is respectively modeled by $\sum_\ell \min\left(\mathbf{P}_{1,i\ell}, \mathbf{P}_{2,\ell j}\right)$ and $\sum_k \boldsymbol{\gamma}_{1,ik} \cdot \boldsymbol{\gamma}_{2,kj}/g_k$.

network simplex method [16] to solve it. Building on prior work [8], we endeavor to accelerate the solution of MLOT using matrix iteration algorithms for GPU acceleration.

To achieve fast computation and obtain an approximate solution, we apply entropic regularization to MLOT. The MLOT-Sinkhorn algorithm is proposed through alternating iterations of scaling variables [8] and intermediate distributions. Theoretical results for our MLOT are also presented, including the global convergence of our MLOT-Sinkhorn algorithm. We first do experiments with a small enough coefficient for entropic regularization. The results show that our MLOT-Sinkhorn can achieve an objective function close to the solution obtained from Gurobi, but tens to hundreds of times faster for larger problems. Furthermore, we view zero-shot retrieval based on CLIP [30] as a transportation problem and utilize MLOT to enhance inference through data augmentation. Specifically, we consider the first layer as features of query images, the second layer as features of the text to be retrieved (i.e., captions), and the third layer as features of the augmented images in the first layer. We employ the MLOT-Sinkhorn algorithm for solving this, and experimental results confirm that this inference method has significantly improved compared to previous softmax-based methods without requiring additional training. Besides, based on the calculation of intermediate distributions, we conducted image interpolation experiments. The results are shown in Appendix A, indicating that the interpolated images generated using MLOT are relatively clear, serving as a viable alternative method for barycentric interpolation. **This paper contributes:**

1) We propose MLOT, where we extend the traditional bipartite graph to a multi-layer structure, in which source marginals transport mass to unknown immediate marginals and then further transport the mass to the target marginal.

2) Entropic regularization is applied to MLOT, and two Sinkhorn-like algorithms for MLOT are derived to obtain an approximate solution for MLOT. We also present theoretical results for the entropic MLOT, which provide estimates of the intermediate distributions and demonstrate the convergence of the entropic MLOT. Experiments demonstrate that under the multi-layer assumption, our method offers a significant advantage in computational efficiency compared to traditional methods.

3) Compared to traditional deep learning tasks, e.g., classification, retrieval, and matching, which mostly rely on a two-layer framework, we propose a Data Augmentation-based learning method that extends the two-layer structure to three layers, where the third layer represents the augmented data from the first layer. We conducted experiments on CLIP-based retrieval and visual graph matching. The results show our new learning framework significantly outperforms the original two-layer case.

## 2  PRELIMINARIES AND RELATED WORK

**Entropic Optimal Transport.** OT dating back to [24], with the objective to seek a mapping that minimizes the total cost of transporting mass from a source measure to a target measure. Kantorovich [19] introduces the idea of using probabilistic transport instead of a deterministic map, which is now commonly known as Kantorovich's OT. Specifically, given the cost matrix $\mathbf{C} \in \mathbb{R}^+_{m \times n}$ and two histograms $(\mathbf{a}, \mathbf{b})$ where $n$ and $m$ are numbers of dimensions, Kantorovich's OT with the entropic regularization [42] involves solving the optimization $\min_{\mathbf{P} \in U(\mathbf{a},\mathbf{b})} \langle \mathbf{C}, \mathbf{P} \rangle - \epsilon H(\mathbf{P})$, where $U(\mathbf{a}, \mathbf{b}) = \{\mathbf{P} \in \mathbb{R}^+_{mn} | \mathbf{P}\mathbf{1}_n = \mathbf{a}, \mathbf{P}^\top \mathbf{1}_m = \mathbf{b}\}$ and $\epsilon > 0$ is the coefficient for entropic regularization $H(\mathbf{P}) = -\langle \mathbf{P}, \log \mathbf{P} - \mathbf{1}_{m \times n} \rangle$. The objective of entropic OT is $\epsilon$-strongly convex,

and thus it has a unique solution, which satisfies $\mathbf{P}_\epsilon^* = \text{diag}(\mathbf{u})\mathbf{K}\text{diag}(\mathbf{v})$, where $\mathbf{K} = e^{-\mathbf{C}/\epsilon}$ is the Gibbs kernel associated to the cost matrix $\mathbf{C}$ and $(\mathbf{u}, \mathbf{v})$ are two (unknown) scaling variables [8].

**Low-Rank Optimal Transport.** Low-rank regularization has been proposed to mitigate the high computational cost and dimensionality issues of classical OT. Recent works [32; 31; 17] directly address the OT problem under a non-negative rank constraint $\mathbf{rk}_+(\mathbf{P}) \leqslant r$ by factorizing the coupling matrix as $P = Q\,\text{diag}(1/g)\,R^\top$, with $Q \in \mathbb{R}^{n \times r}$ and $R \in \mathbb{R}^{m \times r}$, typically solved via Mirror Descent and Dykstra's algorithm. Alternatively, [13] introduces a decomposition through intermediate anchors, where the transport rank is controlled by the number of anchors, and the solution is obtained by alternating optimization of anchor positions and transport matrices. These works essentially focus on constraining the coupling to a multiplicative product structure rather than capturing the semantics of real transport flows. For example, transport from $s_i$ to $t_j$ is recovered by $\sum_k \gamma_{1,ik} \cdot \gamma_{2,kj}/g_k$, which lacks a physical interpretation of flow magnitude. In contrast, our MLOT directly models the actual transported mass, since the global transport can be recovered by $\sum_\ell \min(\mathbf{P}_{1,i\ell}, \mathbf{P}_{2,\ell j})$, highlighting the inherently additive nature of flow.

**Graph Optimal Transport.** The optimal transport on graphs can be traced back to [12], which first calculates the shortest distances between source nodes and target nodes to create a cost matrix, subsequently using it to compute the 1-Wasserstein distance. This approach transforms the problem into a linear program, and more precisely, a min-cost flow problem, which has been utilized and extended to define and study traffic congestion models. Recently, [20] introduced a new variant called Sobolev transport (ST), designed for measures supported on graphs, which allows for a closed-form expression for faster computation. Additionally, [21] generalized Sobolev transport with an Orlicz structure [25]. However, the aforementioned works primarily rely on calculating the shortest distances on the graph, and this simplified graph structure is often difficult to directly apply to deep representation learning. In this paper, we assume the graph structure follows a multi-layered form, and instead of using the shortest path to simplify the graph, we directly compute the inflow and outflow of each node (i.e. intermediate distributions), which can be directly applied to data augmentation-based representation learning.

**Multiple-Marginal Optimal Transport.** Instead of coupling two histograms $(\mathbf{a}, \mathbf{b})$ in Kantorovich problem [19], the multi-marginal optimal transportation [1] couples $K$ histograms $(\mathbf{a}_k)_{k=1}^K$ by solving the following multi-marginal transport:

$$\min_{\mathbf{P} \in \mathbb{R}_{n_1 \times n_2 \dots n_K}^+} \langle \mathbf{C}, \mathbf{P} \rangle = \sum_k \sum_{i_k=1}^{n_k} \mathbf{C}_{i_1, i_2, \dots, i_K} \mathbf{P}_{i_1, i_2, \dots, i_K} \quad \text{s.t.} \sum_{l \neq k} \sum_{i_l=1}^{n_l} \mathbf{P}_{i_1, \dots, i_K} = \mathbf{a}_{k,i_k}, \, \forall k, i_k \quad (1)$$

where $\mathbf{C}_{i_1, i_2, \dots, i_K}$ is $n_1 \times \cdots \times n_K$ cost tensor. Note the Multi-Marginal Optimal Transport has various applications including image processing [29], financial mathematics for derivative pricing [15] and so on [26]. Compared with MLOT, the Multi-Marginal Optimal Transport approach differs in that all of its marginals are deterministic, and its objective is to compute a high-dimensional coupling tensor between multiple marginals, rather than the coupling series between two marginals in this paper.

## 3 MULTI-LAYERED OPTIMAL TRANSPORT

### 3.1 MULTI-LAYERED OPTIMAL TRANSPORT AND ITS ENTROPIC REGULARIZATION

**Formulation of MLOT.** We first give the definition of our Multi-Layered Optimal Transport (MLOT). Given the known source distribution $\mathbf{a}_1$ and target distribution $\mathbf{a}_K$, our MLOT aims to transport the source distribution through intermediate uncertain distributions $(\mathbf{a}_2, \mathbf{a}_3, \dots, \mathbf{a}_{K-1})$ to the target distribution $\mathbf{a}_K$, where $\mathbf{C}_k \in \mathbb{R}_{n_k \times n_{k+1}}^+$ is known as the cost matrix between $\mathbf{a}_k$ and $\mathbf{a}_{k+1}$. Our goal is to solve for the optimal couplings $(\mathbf{P}_k)_{k=1}^{K-1}$ and the intermediate distributions $(\mathbf{a}_k)_{k=2}^{K-1}$ with the following optimization:

$$\min_{(\mathbf{P}_k)_k, (\mathbf{a}_k)_k} \sum_{k=1}^{K-1} \langle \mathbf{C}_k, \mathbf{P}_k \rangle \quad \text{s.t.} \quad \mathbf{P}_k \mathbf{1}_{n_{k+1}} = \mathbf{a}_k, \quad \text{and} \quad \mathbf{P}_k^\top \mathbf{1}_{n_k} = \mathbf{a}_{k+1}, \forall k < K. \quad (2)$$

This formulation is exact an LP, as proved in App. L. Note that when $K = 2$, our MLOT degenerates to the original Kantorovich OT when $\epsilon = 0$. One efficient way to solve the above problem is through

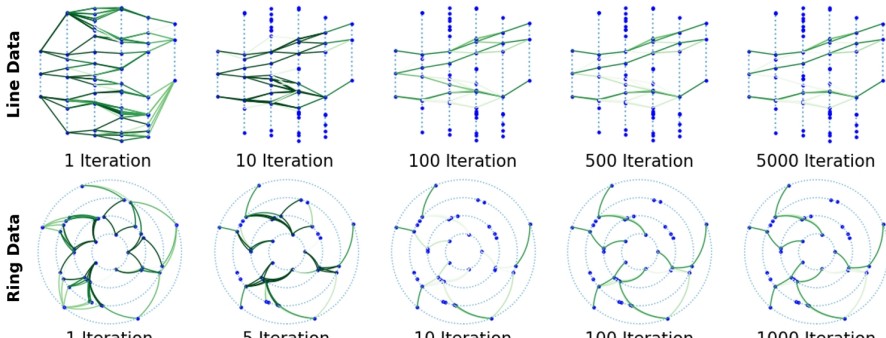

Figure 2: Transportation results of MLOT on synthetic Line and Ring data (refer to the data setup in Sec. 4) and the thickness of the green line is directly proportional to the value of transportation. By varying the iterations, couplings become sharper, and eventually converge to optimal transportation of entropic MLOT.

Graph OT methods based on the shortest path algorithm, as proposed by [34], where the shortest path distances between source and target nodes are first computed, followed by a heuristic algorithm to determine the final solution. However, such algorithms do not directly involve the computation of intermediate distributions $(\mathbf{a}_k)_{k=2}^{K-1}$, limiting their applicability in real-world scenarios. For instance, in the cross-border e-commerce operations problem mentioned in the introduction, if we introduce capacity constraints for goods transportation at ports, which are indeed present in real scenarios and need to be considered, the original shortest path-based algorithms become impractical.

**Enropic Regularization of MLOT.** We then introduce entropy regularization to MLOT in order to obtain a GPU-friendly Sinkhorn-like algorithm, which can iteratively compute an approximate solution for MLOT via matrix iterations. Unlike the case of vanilla OT, MLOT not only requires optimizing coupling $\mathbf{P}_k$ but also involves intermediate distribution $\mathbf{a}_k$. Here, we contemplate applying entropy regularization to both, leading to the formulation of entropic MLOT:

$$\min_{(\mathbf{P}_k)_k, (\mathbf{a}_k)_k} \sum_{k=1}^{K-1} \left( \langle \mathbf{C}_k, \mathbf{P}_k \rangle - \epsilon H(\mathbf{P}_k) \right) - \tau \sum_{k=2}^{K-1} H(\mathbf{a}_k) \ \text{ s.t. } \mathbf{P}_k \mathbf{1}_{n_{k+1}} = \mathbf{a}_k, \ \mathbf{P}_k^\top \mathbf{1}_{n_k} = \mathbf{a}_{k+1}, \forall k. \quad (3)$$

where $\epsilon > 0$ and $\tau \geq 0$ are coefficients for the regularization terms $H(\mathbf{P}_k)$ and $H(\mathbf{a}_k)$, respectively. The optimization described above is essentially a convex optimization problem, ensuring the existence of a unique optimal solution. In particular, as $\epsilon, \tau \to 0$, the entropic MLOT in Eq. 3 degenerates to the original MLOT in Eq. 2. Furthermore, we can further derive properties of the solution as follows by using the method of Lagrange multipliers.

**Proposition 1** (Convergence with $\varepsilon$ and $\tau$). *When regularization on intermediate is canceled ($\tau = 0$), the unique solution $(\mathbf{P}_k^{\tilde{\epsilon},\tau})_k$ of Eq. 3 converges to the optimal solution $\mathbf{P}_k^\star$ of Eq. 2, as $\varepsilon \to 0$:*

$$(\mathbf{P}_k^{\varepsilon,0})_k \xrightarrow{\varepsilon \to 0} \arg\min_{(\mathbf{P}_k)_k} \sum_{k=1}^{K-1} \langle \mathbf{C}_k, \mathbf{P}_k \rangle. \quad (4)$$

*When intermediate is regularized by $\tau$, given fixed $\varepsilon = \varepsilon_0$, the unique solution $(\mathbf{P}_k^{\varepsilon_0,\tau})_k$ of Eq. 3 converges to $(\mathbf{P}_k^{\varepsilon_0,0})_k$ as $\tau \to 0$:*

$$(\mathbf{P}_k^{\varepsilon_0,\tau})_k \xrightarrow{\tau \to 0} \arg\min_{(\mathbf{P}_k)_k} \sum_{k=1}^{K-1} \langle \mathbf{C}_k, \mathbf{P}_k \rangle - \varepsilon_0 H(\mathbf{P}_k). \quad (5)$$

The proof is in Appendix H. Prop. 1 is essentially due to the fact that entropic regularization is a continuous function. This property demonstrates good convergence of MLOT. Eq. 4 and Eq. 5 show respectively that the regularization problem converges to the non-regularization case for both couplings and intermediate. Fig. 10 and Fig. 11 show visually the effect of these two convergences.

### 3.2 TWO ALGORITHMS OF ENTROPIC MLOT AND THEIR CONVERGENCE

In this subsection, we introduce two corresponding Matrix-Scaling-based algorithms: the Bregman iterative algorithm and the Sinkhorn-Knopp algorithm for MLOT.

### 3.2.1 BREGMAN ITERATIVE PROJECTIONS FOR MLOT

We first transform Eq. 3 into an equivalent form of the KL divergence.

**Proposition 2.** *Define the general KL divergence as* $\widetilde{KL}(\mathbf{P}|\mathbf{S}) = \sum_{ij} \mathbf{P}_{ij} \log \frac{\mathbf{P}_{ij}}{\mathbf{S}_{ij}} - \mathbf{P}_{ij} + \mathbf{S}_{ij}$, *the optimization in Eq. 3 is equivalent to the following minimization, where* $(\mathbf{S}_k)_{ij} = e^{-(\mathbf{C}_k)_{ij}/\epsilon}$, *and* $\mathbf{\Delta}_k = \mathbf{1}_{n_k}/n_k$ *represents uniform distribution:*

$$\min_{(\mathbf{P}_k)_k, (\mathbf{a}_k)_k} \varepsilon \sum_{k=1}^{K-1} \widetilde{KL}(\mathbf{P}_k|\mathbf{S}_k) + \tau \sum_{k=2}^{K-1} KL(\mathbf{a}_k|\mathbf{\Delta}_k), \ s.t. \ \mathbf{P}_k \mathbf{1}_{n_{k+1}} = \mathbf{a}_k, \ \mathbf{P}_k^\top \mathbf{1}_{n_k} = \mathbf{a}_{k+1}, \forall k. \quad (6)$$

The proof is given in Appendix G. Prop. 2 shows that the optimal solutions $(\mathbf{P}_k)_k$ and $(\mathbf{a}_k)_k$ exactly minimize the weighted summation of two KL divergences. Then we assume $\tau = 0$ and adopt Bregman projections as proposed in [4] to solve the optimization.

**Bregman Iterations for MLOT.** Following [4], we split the constraints, defining the constraint sets as $\mathcal{C}_{2k-1} = \left\{ \mathbf{P_k} \in \mathbb{R}^{N_k \times N_{k+1}} \mid \mathbf{P_k}^\top \mathbf{1} = \mathbf{a}_k \right\}$ and $\mathcal{C}_{2k} = \left\{ \mathbf{P_k} \in \mathbb{R}^{N_k \times N_{k+1}} \mid \mathbf{P_k} \mathbf{1} = \mathbf{a}_{k+1} \right\}, \forall k.$ Therefore we can find $\mathbf{P_k} \in \mathcal{C}_{2k-1} \cap \mathcal{C}_{2k}, \forall k = 1, .., K-1$. Based on the Bregman projection algorithms, we can iteratively compute $\mathbf{P_k}$, and our improvement lies in the calculation of $\mathbf{a}_k$:

$$\mathrm{Proj}_{\mathcal{C}_{2k-1}}^{KL}(\mathbf{P}_k) = \mathbf{P}_k \, \mathrm{diag}\left(\frac{\mathbf{a}_k}{\mathbf{P}_k^\top \mathbf{1}}\right), \ \mathrm{Proj}_{\mathcal{C}_{2k}}^{KL}(\mathbf{P}_k) = \mathrm{diag}\left(\frac{\mathbf{a}_{k+1}}{\mathbf{P}_k \mathbf{1}}\right) \mathbf{P}_k, \ \mathbf{a}_k = \left((\mathbf{P}_k^\top \mathbf{1}) \odot (\mathbf{P}_{k-1} \mathbf{1})\right)^{1/2}$$
$$(7)$$

The proof is given in AppendixE. Building on [4], we assume that the constraints cycle periodically, i.e., $\mathcal{C}_l = \mathcal{C}_{l+2K}$ for a positive integer index $l < 2K$. The minimization in Eq. 6 can then be solved via the iterative projection scheme as $\mathbf{P}_k^* = \lim_{n \to \infty} \mathrm{Proj}_{\mathcal{C}_n}^{KL}(\mathbf{P}_k^{(l-1)})$ and calculation of $\mathbf{a}_k$ given in Eq. 7 for all $k$. Convergence is guaranteed by the results in [5]. The advantage of this algorithm lies in its simplicity of calculation, requiring no additional variables, making it suitable for training neural networks, and we adopt this algorithm in data augmentation-based Applications in Sec. 3.3. However, its computational efficiency is not optimal. In the following, we propose a more efficient algorithm.

### 3.2.2 MLOT-SINKHORN ALGORITHM AND ITS CONVERGENCE GUARANTEE

**Proposition 3.** *The solution to Eq. 3 is unique, and has the form* $\mathbf{P}_k = diag(\mathbf{u}_k)\mathbf{S}_k diag(\mathbf{v}_k)$ *for* $k = 1, \ldots, K-1$ *where* $\mathbf{S}_k = e^{\mathbf{C}_k/\varepsilon}$, *and* $\{(\mathbf{u}_k, \mathbf{v}_k)\}_k$ *are the set of unknown scaling variables. While the solution of the intermediate distributions satisfying following equations for* $k = 2, 3, \ldots, K-1$:

$$\mathbf{a}_k = \begin{cases} (\mathbf{u}_k \odot \mathbf{v}_{k-1})^{-\epsilon/\tau} & \tau > 0 \\ \left((\mathbf{S}_{k-1}^\top \mathbf{u}_{k-1}) \odot (\mathbf{S}_k \mathbf{v}_k)\right)^{1/2} & \tau = 0 \end{cases} \quad (8)$$

The proof are given in Appendix F. Compared to entropic OT, the coupling form of MLOT is similar, both expressed as the product of the Gibbs kernel $\mathbf{S}_k$ and two diagonal matrices. The difference lies in the fact that our MLOT requires further computation of intermediate distributions as shown in Eq. 8, which implies that the matrix iteration algorithm corresponding to it is inevitably more complex than the Sinkhorn algorithm based on Entropic OT.

**MLOT-Sinkhorn.** Based on Prop. 3, we propose the Sinkhorn-Knopp algorithm for MLOT, which is GPU-friendly and hence accelerates the approximation of the optimal solution of MLOT. the Sinkhorn-like iterative method calculates the optimal solution of Eq. 3 via matrix-vector iterations. To get the results, an intuitive idea is to iteratively update the coupling $\mathbf{P}_k$ and intermediate distributions $\mathbf{a}_k$ until convergence. Thus for updating the coupling $\mathbf{P}_k$, based on the solution form $\mathbf{P}_k = \mathrm{diag}(\mathbf{u}_k)\mathbf{S}\mathrm{diag}(\mathbf{v}_k)$ and the

---

**Algorithm 1:** MLOT-Sinkhorn Algorithm

**Input:** Source distribution $\mathbf{a}_1$, target distribution $\mathbf{a}_K$, distance metrics $(\mathbf{C}_k)_k$, $\varepsilon, \tau$
Initialize $\mathbf{S}_k = \exp(-\mathbf{C}_k/\varepsilon)$, $\mathbf{u}_k = \mathbf{1}$, $\mathbf{v}_k = \mathbf{1}$ for $\forall k < K$ and $\mathbf{a}_k = \mathbf{1}/N_k$ for $\forall 1 < k \leq K$.
**while** not Converge **do**
  **for** $k = 1, 2, \ldots, K-1$ **do**
    $\mathbf{u}_k \leftarrow \mathbf{a}_k \oslash \mathbf{S}_k \mathbf{v}_k$
    $\mathbf{v}_k \leftarrow \mathbf{a}_{k+1} \oslash \mathbf{S}_k^\top \mathbf{u}_k$
    **if** $k > 1$ **then**
      Update $\mathbf{a}_k$ via Eq. 9
    **end if**
  **end for**
**end while**
Calculate $\mathbf{P}_k \leftarrow \mathrm{Diag}(\mathbf{u}_k)\mathbf{S}_k \mathrm{Diag}(\mathbf{v}_k)$ for $\forall k < K$
**Output:** the couplings $(\mathbf{P}_k)_{k=1}^{K-1}$ and the intermediate distributions $(\mathbf{a}_k)_{k=2}^{K-1}$

---

Figure 3: Reformulated procedures of visual graph matching (left) and Image-to-Text retrieval (right).

marginal constraints (i.e. $\mathbf{P}_k \mathbf{1}_{n_{k+1}} = \mathbf{a}_k$ and $\mathbf{P}_k^\top \mathbf{1}_{n_k} = \mathbf{a}_{k+1}$), we derive the following iterations for $\mathbf{u}_k^{(l)}$ and $\mathbf{v}_k^{(l)}$ given the iteration $l$:

$$\mathbf{u}_k^{(l+1)} = \frac{\mathbf{a}_k^{(l)}}{\mathbf{S}_k \mathbf{v}_k^{(l)}}, \ \mathbf{v}_k^{(l+1)} = \frac{\mathbf{a}_{k+1}^{(l)}}{\mathbf{S}_k^\top \mathbf{u}_k^{(l+1)}}, \ \text{where } \mathbf{a}_k^{(l+1)} = \begin{cases} \left(\mathbf{u}_k^{(l+1)} \odot \mathbf{v}_{k-1}^{(l+1)}\right)^{-\epsilon/\tau} & \tau > 0 \\ \left((\mathbf{S}_{k-1}^\top \mathbf{u}_{k-1}^{(l+1)}) \odot (\mathbf{S}_k \mathbf{v}_k^{(l+1)})\right)^{1/2} & \tau = 0 \end{cases}$$

(9)

where initialization is set as $\mathbf{v}_k = \mathbf{1}_{n_k}$ and $\mathbf{a}_k = \mathbf{1}/N_k$. Then, we iteratively update $(\mathbf{u}_k^{(l)}, \mathbf{v}_k^{(l)})$ and $\mathbf{a}_k^{(l+1)}$ for intermediate distributions for all $k$ until convergence. This process allows us to obtain the final solutions $(\mathbf{P}_k)_k$ and $(\mathbf{a}_k)_k$. Note as $\epsilon \to 0$ and $\tau \to 0$ (or $\tau = 0$), empirical evidence demonstrates that the iterative results of our MLOT-Sinkhorn approach closely approximate the exact solution of MLOT obtained using Gurobi.

**Global Convergence of MLOT-Sinkhorn.** The global convergence of MLOT-Sinkhorn is established and greatly simplified with the aid of the Hilbert projective metric $d_{\mathcal{H}}(\mathbf{u}, \mathbf{u}') \overset{\text{def.}}{=} \log \max_{i,j} \frac{\mathbf{u}_i \mathbf{u}'_j}{\mathbf{u}_j \mathbf{u}'_i}$. Several important properties of Hilbert metric are studied in Appendix D.1. For solution form $\mathbf{P}_k = \mathrm{diag}(\mathbf{u}_k) \mathbf{S}_k \mathrm{diag}(\mathbf{v}_k)$ of MLOT-Sinkhorn, the convergence property of $\mathbf{u}_k$ or $\mathbf{v}_k$ is presented as follows.

**Proposition 4** (Convergence for $\tau = 0$). *For all layers, the worst error bound of $\mathbf{u}_k^{l+1}$ is:*

$$d_{\mathcal{H}}\left(\mathbf{u}_k^l, \mathbf{u}_k^*\right) = \mathcal{O}\left[\left(\frac{\gamma^2(\gamma+2)}{2 - 2\gamma^2 - \gamma^3}\right)^l\right], \ \text{where } \gamma = \max_k \lambda(\mathbf{S}_k) \overset{\text{def.}}{=} \sup\left\{\frac{d_{\mathcal{H}}(\mathbf{S}_k \mathbf{y}, \mathbf{S}_k \mathbf{y}')}{d_{\mathcal{H}}(\mathbf{y}, \mathbf{y}')}, \mathbf{y}, \mathbf{y}' \in \mathbb{R}_+^n\right\},$$

(10)

*where $\mathbf{u}^*$ is the unique optimal scaling variable, $\mathbf{u}^l$ is the $l$-th iteration of the scaling variable, and $\lambda(\mathbf{S}_k) \in [0, 1]$ stands for the contraction radio of $\mathbf{S}_k$, which highlights the fact that positive matrix $\mathbf{S}_k$ is a strict contraction on the cone of positive vectors.*

This proposition is proved in Appendix D. The bound for $d_{\mathcal{H}}\left(\mathbf{v}_k^l, \mathbf{v}_k^*\right)$ follows a similar form as $\mathbf{u}_k$. Eq. 10 implies that given proper setting of $\varepsilon, \tau$, the MLOT-Sinkhorn algorithm will perform linear convergence to a $\delta$-approximate solution in $\mathcal{O}(|\log \delta|)$ iterations. Besides, for $\tau > 0$, we also give the convergence results in Appendix D.3.

### 3.3 DATA AUGMENTATION-BASED APPLICATIONS WITH MLOT

We now discuss the application of MLOT to address tasks that involve augmented data, framing the problem through the lens of representation learning theory. Data augmentation is widely adopted in contrastive learning (CL) strategies which typically optimize the InfoNCE-Loss. This loss formulates representation learning as a softmax classification problem, pulling positive pairs together while pushing negative pairs apart. OT-CLIP [33] provides a geometric interpretation of this process, demonstrating that CL can be formulated as a point-set matching problem, where the standard Softmax function is proven to be the optimal solution for this specific Entropic OT problem. However, this traditional bipartite structure limits the model's ability to fuse information from multiple data fields simultaneously, which can derive from augmented data or grouped-stucture in dataset.

Extending the line of [33], we propose using MLOT to generalize this relationship, effectively functioning as a **multiple-layered Softmax**. While vanilla OT (and by extension, standard Softmax) restricts optimization to a two-layer network, MLOT leverages augmented data to formulate a chain-transport problem. By introducing a multi-layered structure, we can contrast multiple positive and negative sample sets at the same time within a single optimization pass. This formulation allows us to integrate multile view as an intrinsic part of the transportation flow.

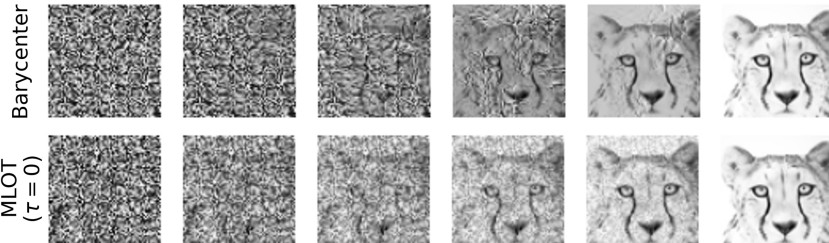

Figure 4: Interpolation Image Computation via Barycenter Calculation and MLOT (K=6). In this example, traditional entropic barycenters require calculating each one individually by varying barycenter weights, and the results are independent of each other. In contrast, our method computes all intermediate results together.

We propose a unified paradigm, illustrated in Fig 3, that jointly considers original data, target data, and augmented data by reformulating the task as an MLOT problem. The fundamental advantage of this formulation is that it shifts the problem definition from "transporting from one **known** distribution to one **unknown** target" to "transporting between **two known** distributions." In tasks like zero-shot retrieval, the true distribution of the retrieved items (the target) is unknown. Standard bipartite methods must assume a uniform prior on the target, which lacks physical significance and limits performance. In contrast, our paradigm constructs a three-layer transport: $U_O \to X_T \to U_{O'}$, where each entry of $U_O, U_{O'}$ represents the original sample and augmented sample. Since they need to be matched once and only once, $U_O, U_{O'}$ is set to uniform distribution. The core improvement lies in treatment towards candidates $X_T$: since the chosen entry is unknown, we hide it into the intermediate layer. This allows the distribution of the choice of retrieved items to be adaptively computed, serving as latent middle distribution ($a_k$). Specifically, we generate cost matrix for any adjacent layers: $C_1$ between $U_O, X_T$, and $C_2$ between $X_T, U_{O'}$.

$$\min_{\mathbf{P}_1 \mathbf{P}_2} \sum_{i=1,2} \langle \mathbf{C}_i, \mathbf{P}_i \rangle - \epsilon H(\mathbf{P}_i) \text{ s.t. } \mathbf{P}_1 \mathbf{1}_{n_2} = \mathbf{1}, \mathbf{P}_1^\top \mathbf{1}_{n_1} = \mathbf{a}_2, \mathbf{P}_2 \mathbf{1}_{n_3} = \mathbf{a}_2, \mathbf{P}_2^\top \mathbf{1}_{n_2} = \mathbf{1}.$$

Here $\mathbf{P}_1$ and $\mathbf{P}_2$ are the two matching score matrices, and $(P_1 + P_2)/2$ is used for overall prediction. Then we explore applications of this paradigm in two downstream tasks that involve data augmentation: graph matching and image-text retrieval. The procedure is discussed in detail below.

**Learning-based Visual Graph Matching.** Graph matching aims at discovering node matching between graphs. Learning-based GM, such as NGMv2[39] and GCAN[18], rely on deep network to construct features solve a bipartite matching problem. One challenge in visual task is **Partial Matching** in the presence of outliers. Several works were done, including traditional algorithm ZACR[37] and learning-based module AFA[40], generally operate within a fixed two-view framework and do not explicitly predict outliers. Our motivation comes from the adaptive middle layers in MLOT, **this unknown distribution is well-mathced with the target image that has unknown inlier distribution**. By generating augmented view of source graph, we formulate a three-layered MLOT problem to solve two matching jointly, and hide outlier distribution into latent layer, as shown in the left part of Fig. 3. The two similarities respectively derived from two GM network (the two network can be either the same or different). This formulation aims to transport all inliers from source image to its augmented twins, and the pass-by middle is exactly the chosen inliers distribution.

**CLIP-based Text-Image retrieval.** Image-Text Retrieval is a traditional multimodal task aimed at establishing correspondence between images and their descriptive text. Zero-shot retrieval, facilitated by models like CLIP[30], aims to retrieve relevant items without any prior training on specific categories or datasets. OT-CLIP[33] proves the insight that traditional bipartite approaches Softmax is equivalent to optimize an OT problem. Note its fundamental limitation that can only leverage two view of samples at single time, we address this by integrating augmented view using MLOT framework, constructing a three-layered flow: Query $\to$ Candidates $\to$ Aug. Query, shown in right part of Fig. 3. Besides the ability to integrate multiple view, this framework also hides unknown retrieved sample distribution into adaptive middle layer. Comparing to the wrong unifrom prior assumption made in Softmax, MLOT formulates the retrieval process in a more accurate way.

**Image Interpolation.** Computing Intermediate Images is a traditional task aimed at generating transitions between two given images, often used for smooth interpolation or data completion. Computing a single intermediate image can be reduced to calculating the (weighted)barycenter between two images. However, if we want a smooth transform path from one image to another, barycenter-based approach [47] requires varying barycenter weights to generate interpolations one

Table 1: **Experiment on synthetic Line and Ring datasets.** The objective and time cost (in seconds) are evaluated by comparing our proposed MLOT-sinkhorn ($\tau = 0$ and $\tau > 0$) with the other two baselines. Our proposed algorithms provide highly accurate results in a much more efficient time.

| | Gurobi | | Short Path+Sinkhorn | | MLOT($\tau = 0$) | | MLOT($\tau > 0$) | |
|---|---|---|---|---|---|---|---|---|
| Size | Obj. | Time(s) | Obj. | Time(s) | Obj. | Time(s) | Obj. | Time(s) |
| | | | Experiment on synthetic Line data. | | | | | |
| 100 | 1.0684 | 0.08 | 1.0692 | 0.17 | 1.0692 | 2.23 | 1.0702 | 1.41 |
| 1K | 0.4082 | 6.64 | 0.4099 | 10.2 | 0.4106 | 2.36 | 0.4126 | 1.61 |
| 2K | 0.6323 | 43.9 | 0.6336 | 13.3 | 0.6342 | 2.90 | 0.6349 | 1.94 |
| 5K | 0.1463 | 330 | 0.1487 | 67.8 | 0.1508 | 11.3 | 0.1519 | 7.40 |
| 10K | Out Of Memory | | 0.3710 | 421 | 0.3707 | 41.2 | 0.3708 | 27.3 |
| 20K | Out Of Memory | | 0.1129 | 2575 | 0.1137 | 162 | 0.1139 | 110 |
| | | | Experiment on synthetic Ring data. | | | | | |
| 100 | 2.3843 | 0.16 | 2.3848 | 0.34 | 2.3874 | 2.97 | 2.3900 | 2.06 |
| 1K | 2.0319 | 20.5 | 2.0341 | 1.24 | 2.0396 | 3.34 | 2.0403 | 2.16 |
| 2K | 2.0402 | 45.6 | 2.0427 | 2.72 | 2.0481 | 3.58 | 2.0484 | 2.27 |
| 4K | 2.0222 | 324 | 2.0249 | 15.3 | 2.0301 | 5.42 | 2.0303 | 3.51 |
| 10K | Out Of Memory | | 2.1536 | 336 | 2.1588 | 47.4 | 2.1589 | 30.2 |
| 20K | Out Of Memory | | 2.1521 | 3125 | 2.1573 | 184 | 2.1573 | 125 |

by one in multiple steps. This task can be naturally formulated as an MLOT problem and thus gain efficiency, since intermediate distributions can be treated as interpolations directly, as shown in Fig.4.

# 4 EXPERIMENTS

## 4.1 EXPERIMENTS ON SYNTHETIC DATA

To validate the efficiency and convergence performance of MLOT-Sinkhorn, particularly with small $\varepsilon, \tau$, we generated synthetic datasets by randomly distributing points in multi-layered structure to simulate MLOT scenario, and conducted extensive numerical experiments.

**Settings.** Scenarios of the MLOT problem were modeled with randomly distributed points. The key information of our synthetic dataset includes: Total number of points $N$(Problem size), Number of layers $K$, number of points per layer $(n_k)_k$, cost metric(measure). The synthetic dataset includes two geometric metric: **Line** problem with $\ell 2$ Euclidean distance, **Ring** problem with Archimedean spiral length metric (see Appendix J). A visualization of the synthetic dataset is shown in Fig. 2, the couplings are initialized as uniform. The thickness of the green line is proportional to the value.

There are two baselines for numerical experiment: (a). Commercial solver Gurobi running on CPUs, (b). GraphOT based method on GPU. The latter baseline convert MLOT into classic OT by firstly computes shortest-path to convert $K - 1$ distance matrices into one direct overall cost matrix. We conduct experiments varying problem size $N$ from $1 \times 10^2$ to $2 \times 10^4$, for both $\tau = 0$ and $\tau > 0$ version, examining the accuracy and running time of the MLOT-Sinkhorn.

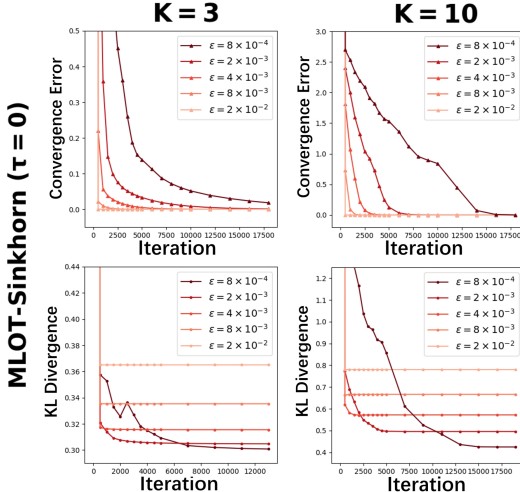

Figure 5: Global convergence and local convergence of MLOT-Sinkhorn ($\tau = 0$). Here present experiments conducted on $K = 3$ and $K = 10$ synthetic dataset. (First row) Numerical changes of $(a_k)_k$ during each iteration. (Second row) KL error between $(a_k)_k$ and ground truth distribution.

The results are shown in Tab. 1. For various problem sizes, MLOT-Sinkhorn has highly consistent objective values with Gurobi, with average relative errors $\sim 0.7\%$. MLOT-Sinkhorn performes several times faster than both Gurobi and GraphOT-based method. As $N$ reaches $1 \times 10^4$, the memory requirements for LP solver become prohibitive, while MLOT-Sinkhorn efficiently handles larger problem sizes while maintaining both high speed and accuracy. Furthermore, Fig. 5 illustrates the convergence performance of MLOT-Sinkhorn with varying iterations.

Table 2: F1(%) on PascalVOC. PMH means Partial Matching Handling. Our method is marked as gray. The score is improved in 14/20 classes with small epochs fine-tuning.

| GM-Network | PMH | aero | bike | bird | boat | bottle | bus | car | cat | chair | cow | table | dog | horse | mbike | person | plant | sheep | sofa | train | tv | mean |
|---|---|---|---|---|---|---|---|---|---|---|---|---|---|---|---|---|---|---|---|---|---|---|
| ZACR[37] | ZACR | 29.87 | 48.70 | 49.10 | 33.85 | 76.59 | 57.03 | 39.12 | 50.37 | 29.16 | 43.80 | 32.83 | 48.68 | 44.25 | 43.28 | 28.95 | 69.09 | 44.84 | 30.16 | 59.41 | 82.38 | 47.07 |
| PCA-GM[38] | None | 37.45 | 59.13 | 50.97 | 37.46 | 78.82 | 65.41 | 44.03 | 52.28 | 33.01 | 48.77 | 38.33 | 53.28 | 48.54 | 50.31 | 34.30 | 78.62 | 50.58 | 31.18 | 64.49 | 85.00 | 52.10 |
| GMN-GM[46] | None | 33.22 | 56.20 | 48.53 | 38.68 | 79.75 | 58.58 | 42.77 | 50.18 | 32.68 | 49.31 | 59.83 | 48.34 | 48.35 | 51.08 | 27.41 | 75.41 | 50.29 | 28.83 | 69.65 | 86.54 | 51.78 |
| CIE[45] | None | 43.08 | 65.84 | 56.30 | 42.26 | 84.03 | 64.25 | 44.56 | 57.11 | 34.24 | 55.50 | 48.83 | 57.31 | 54.02 | 57.22 | 34.68 | 84.94 | 53.24 | 41.57 | 68.00 | 86.61 | 56.68 |
| NGMv2[39] AFA-I[40] | AFA-I[40] | 51.53 | 69.24 | 67.91 | 57.52 | 90.42 | 76.95 | 62.92 | 66.68 | 47.29 | 66.08 | 52.67 | 66.08 | 62.62 | 68.77 | 49.47 | 96.63 | 61.16 | 42.75 | 90.22 | 87.85 | 66.74 |
| | AFA-U[40] | 50.61 | 68.04 | 66.39 | 53.92 | 89.83 | 76.31 | 61.31 | 66.06 | 45.34 | 65.12 | 60.25 | 64.71 | 61.06 | 68.13 | 48.76 | 95.56 | 61.04 | 44.09 | 90.02 | 88.50 | 66.25 |
| | MLOT(ours) | 52.16 | 67.50 | 69.73 | 58.93 | 90.35 | 79.44 | 69.03 | 67.82 | 47.29 | 69.41 | 54.83 | 67.99 | 64.16 | 68.30 | 51.73 | 96.85 | 64.43 | 41.19 | 90.59 | 87.30 | 67.95 |
| GCAN[18] | AFA-I | 51.49 | 71.09 | 67.98 | 55.95 | 90.96 | 78.76 | 61.47 | 68.37 | 52.71 | 69.94 | 60.00 | 68.62 | 66.62 | 69.93 | 49.34 | 97.57 | 64.15 | 51.27 | 89.67 | 89.49 | 68.77 |
| | AFA-U | 51.99 | 71.47 | 68.49 | 55.13 | 91.04 | 78.03 | 62.30 | 68.39 | 53.89 | 69.95 | 57.50 | 68.19 | 66.04 | 70.61 | 49.62 | 97.49 | 63.54 | 58.57 | 89.28 | 89.89 | 68.97 |
| | MLOT(ours) | 50.90 | 70.00 | 70.40 | 60.01 | 91.61 | 79.06 | 65.57 | 68.43 | 52.78 | 71.47 | 60.83 | 69.19 | 67.35 | 70.53 | 52.04 | 97.19 | 65.27 | 50.42 | 92.20 | 88.56 | 69.69 |

Table 3: CLIP-based Zero shot Image-Text retrieval on COCO and Flickr, with random geometric transformation on images and random selection on captions.

| Inference | COCO | | | | | | Flickr30k | | | | | |
|---|---|---|---|---|---|---|---|---|---|---|---|---|
| | Image⇒Text | | | Text⇒Image | | | Image⇒Text | | | Text⇒Image | | |
| | R@1 | R@5 | R@10 | R@1 | R@5 | R@10 | R@1 | R@5 | R@10 | R@1 | R@5 | R@10 |
| ViT-B/32 structure | | | | | | | | | | | | |
| Softmax | 49.8 | 74.6 | 83.1 | 29.0 | 52.8 | 64.3 | 34.3 | 54.4 | 62.0 | 24.4 | 43.0 | 51.0 |
| Independent Sinkhorn | 46.4 | 71.5 | 79.8 | 32.1 | 58.0 | 68.5 | 36.4 | 60.0 | 69.6 | 24.8 | 45.1 | 54.9 |
| **MLOT(Rand. Augmentation)** | 50.7 | 75.1 | 83.3 | 35.1 | 61.2 | 72.2 | 41.0 | 65.3 | 74.3 | 27.4 | 50.0 | 59.8 |
| RN50x64 structure | | | | | | | | | | | | |
| Softmax | 57.4 | 80.6 | 88.0 | 35.6 | 60.2 | 70.1 | 45.1 | 65.3 | 71.7 | 33.1 | 52.6 | 60.0 |
| Independent Sinkhorn | 56.3 | 78.9 | 86.4 | 39.1 | 64.7 | 74.4 | 51.0 | 74.9 | 82.6 | 35.1 | 57.5 | 66.6 |
| **MLOT(Rand. Augmentation)** | 58.0 | 81.1 | 88.1 | 43.1 | 70.3 | 79.6 | 54.0 | 77.4 | 84.6 | 41.6 | 65.5 | 74.7 |

## 4.2 EXPERIMENTS ON DATA AUGMENTATION-BASED LEARNING

**Experiments on CLIP-based Text-Image retrieval.** For downstream task image-text retrieval, we use COCO2017 [22] 5k validation set and Flickr [44] 30k dateset. Two different structures of the CLIP model (ViT-B/32, RN50x64) are used to compute the feature embedding of images and texts. The widely-used $R@m(m = 1, 5, 10)$ in cross-modal retrieval is reported for performance evaluation. The baseline uses Softmax or vanilla OT-Sinkhorn to predict retrieval image (or text) solely based on information from bipartite structure. By integrating augmented data information via MLOT framework shown in Fig. 3, we obtain significant improvement in recall of zero-shot retrieval on both datasets and tasks. As shown in Tab. 3, the recall rate is improved by 4.2% for both Transformer and ResNet architecture on average compared to vanilla OT.

**Experiments on Visual Graph Matching.** Following [41], we conduct the partial visual graph matching experiment on PascalVOC [11] with outlier setting: Given image $\mathcal{S}$ without outliers, and image $\mathcal{T}$ with outliers, the task is to detect all outliers as well as predict precise matching. The baseline includes severeal GM methods mentioned in Sec. 3.3. The Partial Matching Handling (PMG) refers to post-method to realize partial match. Following the procedure proposed in Fig. 3, we integrate information from $\mathcal{S}_{augment}$ via MLOT framework. Thus reformulate the problem into transporting distribution of $\mathcal{S}$ (uniform) to distribution of $\mathcal{S}_{augment}$ (uniform) and viaway intermediate $\mathcal{T}$ (inliers distribution to be predicted). Based on this MLOT framework, we fine-tune 5 epochs on NGMv2 and GCAN networks. The average F1-score on entire classes is reported in Tab. 2. MLOT framework presents improvement in 14/20 classes with fine-tuning by leveraging augmented information.

**More Experiments on Computing Image Interpolation.** Note that our MLOT can be used to efficiently compute any number of interpolation images between two given images. Fig. 4 shows the results between two $64 \times 64$ grayscale image. In contrast to traditional barycenter-based methods [47], require calculating each one individually by varying barycenter weights, the intermediate layers in MLOT automatically represent interpolation. More details are given in Appendix A. We also compute such morphing process on CelebA [23], a high-resolution $218 \times 178$ human-face coloured image datasets. The results are shown in Fig. 7.

## 5 CONCLUSION AND LIMITIONS

In this paper, we have proposed Multi-layered Optimal Transport (MLOT), a novel approach extending traditional optimal transport to handle complex, multi-stage transportation scenarios. We then introduce the MLOT-Sinkhorn algorithms, leveraging entropic regularization for efficient computation on GPUs. However, our algorithm relies on the prior hierarchical structure, thus cannot deal with more general graphs, which are the areas that require further investigation.

## 6 ETHICS STATEMENT

This work adheres to the ICLR Code of Ethics. Our study uses only publicly available datasets, without involving sensitive information. We do not anticipate major ethical risks, though we encourage responsible use of the proposed methods.

## 7 REPRODUCIBILITY STATEMENT

We provide implementation details, hyperparameters, and dataset descriptions in the main text and appendix. The dataset is either publicly accessible or can be fabricated through the code we provide, and we include sufficient information to reproduce the reported results. Source code and scripts will be released to ensure full reproducibility.

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

## A  Visual Experiments on intermediate distributions

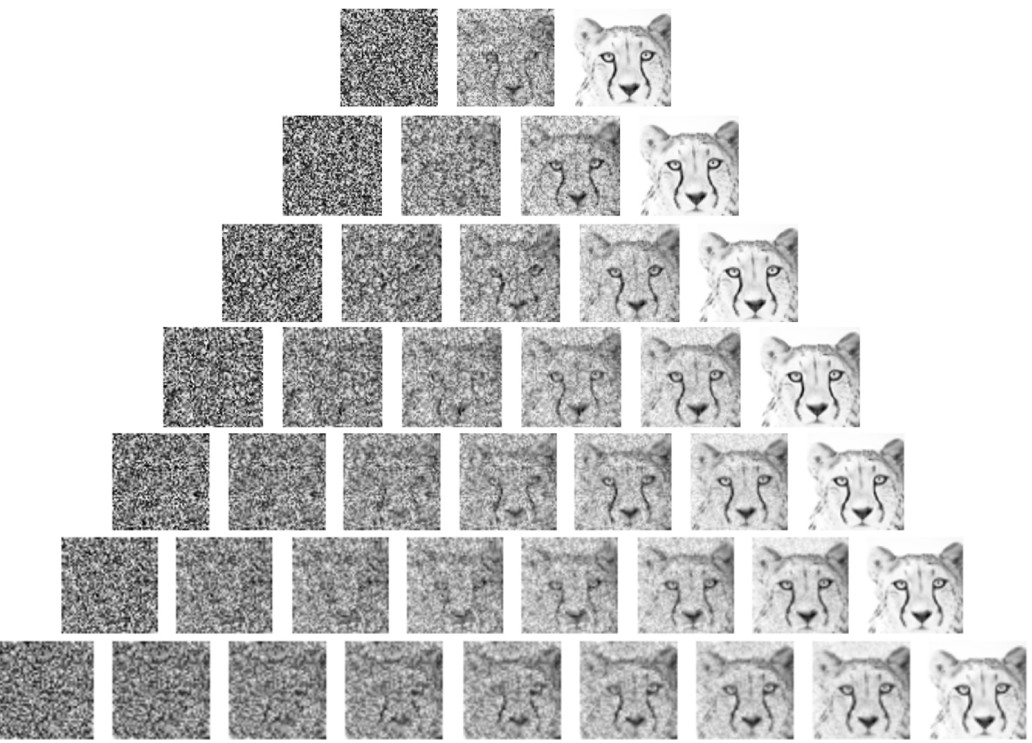

Figure 6: Intermediate images between given picture ($64 \times 64$, grayscale), generated by MLOT ($\varepsilon = 1 \times 10^{-5}, \tau = 0$). Each row represents reformulating as MLOT with different layer amount ($K = 3$ to $K = 9$). The layers $(\mathbf{a}_k)_k$ in MLOT are regarded as grayscale distribution of intermediate images. Results demonstrate the effectiveness and smooth transformation of images under MLOT framework. Layers in different location are equivalent to different setting of $\lambda$ in barycenter method.

**Relation to Wasserstein Barycenter.** We found that our MLOT can be linked to the Wasserstein barycenter. For the distributions $(\mathbf{b}_s)_{s=1}^S$, the Wasserstein barycenter among them aims to learn the distribution $\mathbf{a}$:

$$\min_{(\mathbf{P}_s)_s, \mathbf{s}} \sum_{s=1}^S \lambda_s < \mathbf{D}_s, \mathbf{P}_s > \quad \text{s.t.} \quad \mathbf{P}_s \mathbf{1} = \mathbf{b}_s, \quad \mathbf{P}_s \mathbf{1} = \mathbf{a} \quad \forall s = 1, 2, \ldots, S \quad (11)$$

where $\mathbf{D}_s$ is the distance matrix between $\mathbf{a}$ and $\mathbf{b}_s$. As mentioned in MLOT formulation, our MLOT assumes that the source and target distributions are known, and the objective is to compute the intermediate distributions. In contrast, the Wasserstein barycenter assumes that one or several target distributions of the transportation are known, and the goal is to compute the source distribution. Specifically, when $S = 2$ in Eq.11 and $K = 3$ in Eq.2, the optimization of our MLOT is equivalent to solving the Wasserstein barycenter by setting $\mathbf{C}_1 = \mathbf{D}_1^\top$ and $\mathbf{C}_2 = \mathbf{D}_2$. In this paper, following [8], we consider MLOT under entropic regularization in the next subsection, where we directly compute

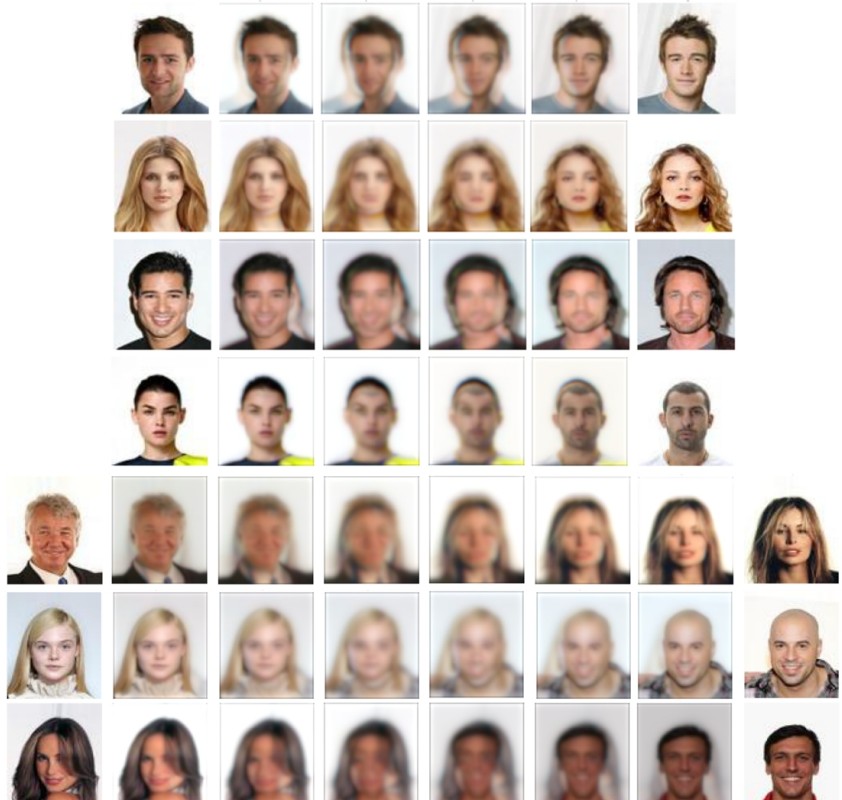

Figure 7: Computing interpolation image on CelebA high-resolution dataset ($218 \times 178$, coloured). $K = 6, 8$ layers.

the coupling between each pair of layers and intermediate distributions instead of relying on indirect calculations through shortest paths.

As mentioned above, this task is mostly addressed by calculating the barycenter of two given images, where different weights are set to generate a coherent series of intermediate images.

Sepcifically, given two 64x64 grayscale image $\mathbf{f}_S, \mathbf{f}_T$, a typical solution is to compute their barycenter. The cost metric $D_s, D_t$ is determined by the distances between pixel locations, i.e. pixel-wise Euclidean distance $D$ between two 64x64 grid. Thus the intermediate image $\mathbf{f}$ can be computed under $D_s = \lambda D$ and $D_t = (1 - \lambda)D$. By adjusting the metric weight $\lambda$, the resulting intermediate image can be biased to varying degrees.

However, if several intermediate images are required, this barycenter-based method requires recalculating for each barycenter weight. In contrast, if we view all intermediate images with different bias as part of a complete transmission process, we can obtain them within single computation by reformulating the problem as MLOT.

For example, if an intermediate image with rational weight $D_s = \lambda D$, $\lambda = \frac{p}{q}$, $\gcd(p, q) = 1$ is required, we can formulate a MLOT with $K = q + 1$, and all cost metric is set to $D$. Then the $q$-th layer can be regarded as the required image distribution.

Generally, if $\lambda_1, \lambda_2, \ldots, \lambda_k$ weights images are required, barycenter-based method has to compute $k$ times. In contrast, we can formulate it as a MLOT problem. Suppose $\lambda_i = \frac{p_i}{q_i}$ and $\gcd(p_i, q_i) = 1$. Then we can set $K = \mathrm{lcm}(q_1, q_2, \ldots, q_k) + 1$, and all cost metric are set to $D$.

We conducted tests on grayscale images (a random Gaussian noise and a leopard), each sized 64x64. As shown in Fig. 6. MLOT was applied varying $K = 3$ to $K = 9$ layers respectively. The results indicate our proposed method is effective, that the **intermediate layers can be smoothly interpreted as intermediate images**. What is more, MLOT generates several intermediate images **at a single calculation**, which outperforms the barycenter-based method with respect to efficiency.

## B  CLUSTERING-BASED CONTRASTIVE LEARNING VIA MLOT-SINKHORN.

Contrastive learning is an efficient self-supervised learning method that aims to learn features by contrasting positive and negative pairs. [7] employs an online clustering approach for contrastive learning. More precisely, we compute a code from an augmented version of the image and predict this code from other augmented versions of the same image. Given two image features $z_t$ and $z_s$ from two different augmentations of the same image, we compute their codes $q_t$ and $q_s$ by matching these features to a set of $K$ prototypes $\{c_1, \ldots, c_K\}$. We then set up a "swapped" prediction problem with the following loss:

$$L(z_t, z_s) = \ell(z_t, q_s) + \ell(z_s, q_t), \tag{12}$$

which consists of two terms that define the "swapped" prediction problem: predicting the code $q_t$ from the feature $z_s$, and $q_s$ from $z_t$. Each term denotes the cross-entropy loss between the code and probability obtained by applying the Softmax to the dot products of $z_t$ and all prototypes:

$$\ell(z_t, q_s) = -\sum_k q_s^{(k)} \log p_t^{(k)} \text{ s.t. } p_t^{(k)} = \frac{\exp\left(\frac{z_t^\top c_k}{\tau}\right)}{\sum_{k'} \exp\left(\frac{z_t^\top c_{k'}}{\tau}\right)}.$$

For the calculation of $q_s$ and $q_t$, SwAV [7] uses the Sinkhorn algorithm to obtain two matching probability matrices. However, it assumes that all prototypes share a uniform distribution, which is somewhat unreasonable, as the number of samples in each cluster may differ. Instead, we relax the uniform assumption and use MLOT-Sinkhorn to compute the matching for the three-layer features-prototype-features matching result.

## C  HANDLE CONSTRAINTS ON INTERMEDIATE DISTRIBUTION

Suppose there exists a set of additional constraints on intermediate distribution, i.e. $(\mathbf{c}_k)_k$, and $\forall k = 2, ..., K-1$, the constraint forces $\mathbf{a}_k \leqslant \mathbf{c}_k$.

Such situation is especially common in real-world scenario, where warehouse or factories may have storage capacity. Therefore it's crucial to take distribution constraints into consideration.

Our MLOT-Sinkhorn can naturally adapt to these situations, by simply adding clip-function after each update of $\mathbf{a}_k$.

$$\mathbf{a}_k^{(l+1)} = \begin{cases} \min\left[\left(\mathbf{u}_k^{(l+1)} \odot \mathbf{v}_{k-1}^{(l+1)}\right)^{-\epsilon/\tau}, \mathbf{c}_k\right] & \tau > 0 \\ \min\left[\left((\mathbf{S}_{k-1}^\top \mathbf{u}_{k-1}^{(l+1)}) \odot (\mathbf{S}_k \mathbf{v}_k^{(l+1)})\right)^{1/2}, \mathbf{c}_k\right] & \tau = 0 \end{cases} \tag{13}$$

We generate different level of constraints in $K = 3$ MLOT-Sinkhorn experiment, and compare the objective and intermediate distribution with Gurobi ground truth, as shown in Fig. 8

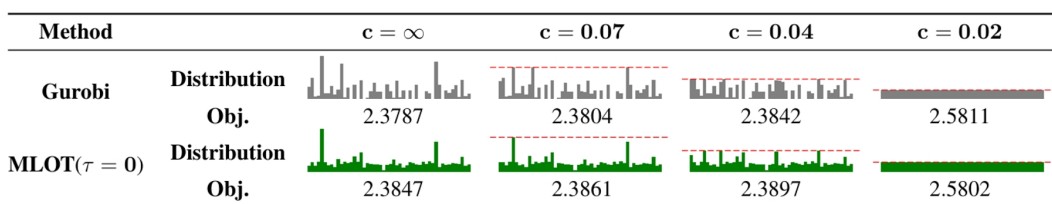

| Method | | $\mathbf{c} = \infty$ | $\mathbf{c} = 0.07$ | $\mathbf{c} = 0.04$ | $\mathbf{c} = 0.02$ |
|---|---|---|---|---|---|
| **Gurobi** | Distribution | | | | |
| | Obj. | 2.3787 | 2.3804 | 2.3842 | 2.5811 |
| **MLOT($\tau = 0$)** | Distribution | | | | |
| | Obj. | 2.3847 | 2.3861 | 2.3897 | 2.5802 |

Figure 8: Setting different constraints on layers.

## D  GLOBAL CONVERGENCE OF MLOT-SINKHORN

### D.1  PROPERTY OF HILBERT METRIC

To measure the gap between iterative result and optimal coupling, Hilbert metric is introduced. $d_{\mathcal{H}}(\mathbf{u}, \mathbf{u}') := \log \max_{i,j} \frac{\mathbf{u}_i \mathbf{u}_j'}{\mathbf{u}_j \mathbf{u}_i'}$. Firstly, several mathematical properties of Hilbert Metric are studied as follow.

1. $d_{\mathcal{H}}\left(\frac{\mathbf{a}}{\mathbf{b}}, \frac{\mathbf{c}}{\mathbf{d}}\right) = d_{\mathcal{H}}(\mathbf{ad}, \mathbf{bc}) \leqslant d_{\mathcal{H}}(\mathbf{a}, \mathbf{c}) + d_{\mathcal{H}}(\mathbf{b}, \mathbf{d})$

   **Proof**: By definition:

   $$LHS = \log \max \frac{\mathbf{a}_i \mathbf{c}_j \cdot \mathbf{b}_j \mathbf{d}_i}{\mathbf{b}_i \mathbf{d}_j \cdot \mathbf{a}_j \mathbf{c}_i} = d_{\mathcal{H}}(\mathbf{ad}, \mathbf{cb})$$

   Separating the product, we have:

   $$LHS \leqslant \log \max \frac{\mathbf{a}_i \mathbf{c}_j}{\mathbf{a}_j \mathbf{c}_i} + \log \max \frac{\mathbf{b}_j \mathbf{d}_i}{\mathbf{b}_i \mathbf{d}_j} = d_{\mathcal{H}}(\mathbf{a}, \mathbf{c}) + d_{\mathcal{H}}(\mathbf{b}, \mathbf{d})$$

2. $d_{\mathcal{H}}(\mathbf{a}^\varepsilon, \mathbf{b}^\varepsilon) = |\varepsilon| d_{\mathcal{H}}(\mathbf{a}, \mathbf{b})$

   **Proof**: By definition: $LHS = \log \max \frac{\mathbf{a}_i^\varepsilon \mathbf{b}_j^\varepsilon}{\mathbf{a}_j^\varepsilon \mathbf{b}_i^\varepsilon}$. Since the operation is to maximize for all $i, j$, whether $\varepsilon > 0$ or $\varepsilon < 0$ will obtain the maximum or minimum at same row/column combination. Therefore the exponent can be separated out as absolute value.

3. $d_{\mathcal{H}}(\mathbf{ta}, \mathbf{tb}) = d_{\mathcal{H}}(t\mathbf{a}, t\mathbf{b})$

   **Proof**: If $t \in \mathbb{R}_+^n$ and $a, b \in \mathbb{R}_+^{n \times m}$. Then expand the by definition will prove this property straight forward. If $t \in \mathbb{R}_+^{w \times n}$, the situation becomes more complicated, which we will discuss immediately below.

## D.2 INTRODUCTION OF CONTRACTION RADIO

In the solution form $\operatorname{diag}(\mathbf{u}_k) \mathbf{S}_k \operatorname{diag}(\mathbf{v}_k)$, the constant argument $\mathbf{S}_k$ is critical in the convergence process. [28] points out how matrix production influences Hilbert metric. [14] generalizes this as a nature of a matrix, which can be regraded as contraction radio during iteration. As the following proposition shows.

$$d_{\mathcal{H}}(\mathbf{Sv}, \mathbf{Sv}') \leq \lambda(\mathbf{S}) d_{\mathcal{H}}(\mathbf{v}, \mathbf{v}')$$

, where $\lambda(\mathbf{S}) = \frac{\sqrt{\eta(\mathbf{S})} - 1}{\sqrt{\eta(\mathbf{S})} + 1}$ and $\eta(\mathbf{S}) := \max\limits_{ijkl} \frac{\mathbf{S}_{ik} \mathbf{S}_{jl}}{\mathbf{S}_{jk} \mathbf{S}_{il}}$

The $\lambda(\mathbf{S})$ here is defined as

$$\sup \left\{ \frac{d_{\mathcal{H}}(\mathbf{Sy}, \mathbf{Sy}')}{d_{\mathcal{H}}(\mathbf{y}, \mathbf{y}')} , \; \mathbf{y}, \mathbf{y}' \in \mathbb{R}_+^n \right\}$$

, aiming to extract constant from Hilbert metric. Notice that $\lambda(\mathbf{S})$ is larger than 0 and less than 1, we call it contraction radio, denoted as $\gamma$.

## D.3 PROOF OF CONVERGENCE

**The case $\tau > 0$**

Iteration steps (considering $l$-th iteration):

$$\mathbf{u}_k^{l+1} = \mathbf{a}_k^l \oslash \mathbf{S}_k \mathbf{v}_k^l \tag{14}$$

$$\mathbf{v}_k^{l+1} = \mathbf{a}_k^l \oslash \mathbf{S}_k^\top \mathbf{u}_k^l \tag{15}$$

$$\mathbf{a}_k^{l+1} = \left( \mathbf{u}_k^{l+1} \odot \mathbf{v}_{k-1}^{l+1} \right)^{-\epsilon/\tau} \tag{16}$$

Denote the optimal value as $\mathbf{u}_k^*, \mathbf{v}_k^*, \mathbf{a}_k^*$. Now consider the Hilbert distance between $l+1$-th iteration to the optimal value:

$$d_{\mathcal{H}}(\mathbf{u}^{l+1}, \mathbf{u}^*) = d_{\mathcal{H}}\left(\frac{\mathbf{a}^l}{\mathbf{S}\mathbf{v}^l}, \frac{\mathbf{a}^*}{\mathbf{S}\mathbf{v}^*}\right) \tag{17}$$

$$\leqslant \lambda(\mathbf{S})\left[d_{\mathcal{H}}\left(\mathbf{a}^l, \mathbf{a}^*\right) + d_{\mathcal{H}}\left(\mathbf{v}^l, \mathbf{v}^*\right)\right] \tag{18}$$

$$d_{\mathcal{H}}\left(\mathbf{v}^{l+1}, \mathbf{v}^*\right) = d_{\mathcal{H}}\left(\frac{\mathbf{a}^l}{\mathbf{S}^\top \mathbf{u}^l}, \frac{\mathbf{a}^*}{\mathbf{S}^\top \mathbf{u}^*}\right) \tag{19}$$

$$\leqslant \lambda(\mathbf{S})\left[d_{\mathcal{H}}\left(\mathbf{a}^l, \mathbf{a}^*\right) + d_{\mathcal{H}}\left(\mathbf{u}^l \mathbf{u}^*\right)\right] \tag{20}$$

$$d_{\mathcal{H}}\left(\mathbf{a}^l, \mathbf{a}^*\right) = d_{\mathcal{H}}\left(\left(\mathbf{u}^l \odot \mathbf{v}^l\right)^{-\frac{\varepsilon}{\tau}}, \left(\mathbf{u}^* \odot \mathbf{v}^*\right)^{-\frac{\varepsilon}{\tau}}\right) \tag{21}$$

$$\leqslant \frac{\varepsilon}{\tau}\left[d_{\mathcal{H}}\left(\mathbf{u}^l, \mathbf{u}^*\right) + d_{\mathcal{H}}\left(\mathbf{v}^l, \mathbf{v}^*\right)\right] \tag{22}$$

The layer number $k$ is not important here, since we can simply replace all $\mathbf{a}_k^l, \mathbf{u}_k^l, \mathbf{v}_k^l, \gamma_k$ by the biggest one in this iteration, which guarantee a worst bound.

Substitute Eq. 20 into Eq. 22, we have:

$$d_{\mathcal{H}}\left(\mathbf{a}^l, \mathbf{a}^*\right) \leqslant \frac{\varepsilon}{\tau}\frac{1+\gamma}{1-(\varepsilon/\tau)\gamma} \cdot d_{\mathcal{H}}\left(\mathbf{u}^l, \mathbf{u}^*\right)$$

Substitute this into Eq. 18, finally we have:

$$d_{\mathcal{H}}\left(\mathbf{u}^{l+1}, \mathbf{u}^*\right) \leqslant \frac{\gamma}{1-(\varepsilon/\tau)\gamma}\left(\gamma + \frac{2\varepsilon}{\tau}\gamma + \frac{\varepsilon}{\tau}\right) \cdot d_{\mathcal{H}}\left(\mathbf{u}^l, \mathbf{u}^*\right)$$

Which indicates the Hilbert difference between $\mathbf{u}^l$ and optimal $\mathbf{u}^*$ converges in a exponential speed.

$$d_{\mathcal{H}}\left(\mathbf{u}^{l+1}, \mathbf{u}^*\right) = \mathcal{O}\left[\left(\frac{\gamma}{1-(\varepsilon/\tau)\gamma}\left(\gamma + \frac{2\varepsilon}{\tau}\gamma + \frac{\varepsilon}{\tau}\right)\right)^l\right]$$

Since the contraction radio $\gamma$ is less than 1 (What's more, in experiment we find that $\gamma$ is always around $0.5\tilde{\,}0.7$), and $\varepsilon/\tau$ is always set less than 0.5, then $d_{\mathcal{H}}\left(\mathbf{u}^{l+1}, \mathbf{u}^*\right) \to 0$.

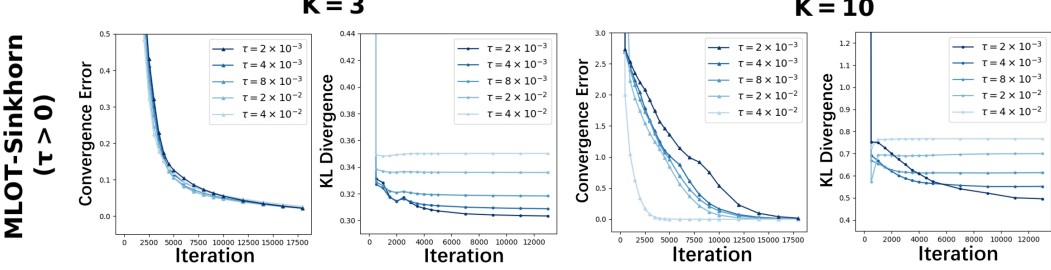

Figure 9: Convergence of MLOT-Sinkhorn ($\tau > 0$), conducted on $K = 3, 10$ synthetic dataset. (First row) Numerical changes of $(a_k)_k$ during each iteration. (Second row) KL error between $(a_k)_k$ and ground truth distribution.

**The case $\tau = 0$**

Iteration steps (considering $l$-th iteration):

$$\mathbf{u}_k^{l+1} = \mathbf{a}_k^l \oslash \mathbf{S}_k \mathbf{v}_k^l$$
$$\mathbf{v}_k^{l+1} = \mathbf{a}_k^l \oslash \mathbf{S}_k^\top \mathbf{u}_k^l \tag{23}$$
$$\mathbf{a}_k^{l+1} = \left(\left(\mathbf{S}_{k-1}^\top \mathbf{u}_{k-1}^{l+1}\right) \odot \left(\mathbf{S}_k \mathbf{v}_k^{l+1}\right)\right)^{1/2}$$

The remain proof is similar as the case $\tau > 0$.

$$d_{\mathcal{H}}\left(\mathbf{a}^l, \mathbf{a}^*\right) \leqslant \frac{1}{2}\gamma_{k-1} d_{\mathcal{H}}\left(\mathbf{u}_{k-1}^{l+1}, \mathbf{u}_{k-1}^*\right) + \frac{1}{2}\gamma_k d_{\mathcal{H}}\left(\mathbf{v}_k^{l+1}, \mathbf{v}_k^*\right)$$
$$\leqslant \frac{1}{2}\gamma d_{\mathcal{H}}\left(\mathbf{u}^{l+1}\right) + \frac{1}{2}\gamma d_{\mathcal{H}}\left(\mathbf{v}^{l+1}\right) \tag{24}$$

, in which we denote $\max\limits_{k} \gamma_k$ as $\gamma$, and represent all layer's Hilbert distance by the biggest one in this iteration $d_{\mathcal{H}}\left(\mathbf{a}^l, \mathbf{a}^*\right)$, etc. We have:

$$(2 - 2\gamma^2 - \gamma^3)d_{\mathcal{H}}\left(\mathbf{a}^l, \mathbf{a}^*\right) \leqslant \gamma^2(1+\gamma)d_{\mathcal{H}}\left(\mathbf{u}^l, \mathbf{u}^*\right) \tag{25}$$

Combine Eq. 18, Eq. 20 and Eq. 25, finally we have:

$$d_{\mathcal{H}}\left(\mathbf{u}^{l+1}, \mathbf{u}^*\right) \leqslant \frac{\gamma^2(\gamma+2)}{2 - 2\gamma^2 - \gamma^3} \cdot d_{\mathcal{H}}\left(\mathbf{u}^l, \mathbf{u}^*\right)$$

Which indicates the Hilbert distance between $\mathbf{u}^l$ and optimal $\mathbf{u}^*$ converges in a exponential speed.

$$d_{\mathcal{H}}\left(\mathbf{u}^{l+1}, \mathbf{u}^*\right) = \mathcal{O}\left[\left(\frac{\gamma^2(\gamma+2)}{2 - 2\gamma^2 - \gamma^3}\right)^l\right]$$

## E    PROOF OF BREGMAN ITERATIONS ALGORITHM FOR MLOT

---

**Algorithm 2:** MLOT-Sinkhorn Algorithm

---

**Input:** Source distribution $\mathbf{a}_1$, target distribution $\mathbf{a}_K$, distance metrics $(\mathbf{C}_k)_k, \varepsilon, \tau$
Initialize $\mathbf{P}_k = \exp(-\mathbf{C}_k/\varepsilon)$ for $\forall k < K$ and $\mathbf{a}_k = \mathbf{1}/N_k$ for $\forall 1 < k < K$.
**while** not Converge **do**
 **for** $k = 1, 2, \ldots, K-1$ **do**
  $\mathbf{P}_k \leftarrow \mathbf{P}_k \operatorname{diag}\left(\frac{\mathbf{a}_k}{\mathbf{P}_k^\top \mathbf{1}}\right)$
  $\mathbf{P}_{k+1} \leftarrow \operatorname{diag}\left(\frac{\mathbf{a}_{k+1}}{\mathbf{P}_k \mathbf{1}}\right) \mathbf{P}_k$
  **if** $k > 1$ **then**
   Update $\mathbf{a}_k \leftarrow \left((\mathbf{P}_k^\top \mathbf{1}) \odot (\mathbf{P}_{k-1}\mathbf{1})\right)^{1/2}$
  **end if**
 **end for**
**end while**
**Output:** the couplings $(\mathbf{P}_k)_{k=1}^{K-1}$ and the intermediate distributions $(\mathbf{a}_k)_{k=2}^{K-1}$

---

Based on KL form of MLOT in Prop. 2, we prove the Bregman iteration algorithm proposed in Eq. 7, We decompose the constraint set as $\forall k = 1, .., K-1, \mathbf{P_k} \in \mathcal{C}_{2k-1} \cap \mathcal{C}_{2k}$, where $\mathcal{C}_{2k-1} = \left\{\mathbf{P_k} \in \mathbb{R}^{N_k \times N_{k+1}} \mid \mathbf{P_k}^\top \mathbf{1} = \mathbf{a}_{k+1}\right\}$ and $\mathcal{C}_{2k} = \left\{\mathbf{P_k} \in \mathbb{R}^{N_k \times N_{k+1}} \mid \mathbf{P_k}\mathbf{1} = \mathbf{a}_k\right\}$.

Firstly we derive the Bregman projection on $\mathcal{C}_{2k-1}$.

Denote $\mathbf{P}_k$ as the projection on $\mathcal{C}_{2k-1}$ of $\hat{\mathbf{P}}_k$. The first-order conditions of $\operatorname{Proj}_{\mathcal{C}_{2k-1}}^{KL}(\hat{\mathbf{P}}_k)$ states the existence of Lagrange multipliers $\mathbf{g}_k$ such that:

$$\varepsilon \log\frac{\mathbf{P}_k}{\hat{\mathbf{P}}_k} + \mathbf{1}^\top \mathbf{g}_k = 0$$

Denote $\mathbf{v}_k = e^{-\mathbf{g}_k/\varepsilon}$. Condition $\mathbf{P_k}^\top \mathbf{1} = \mathbf{a}_{k+1}$ thus implies that

$$\mathbf{v}_k = \frac{\mathbf{a}_{k+1}}{\hat{\mathbf{P}}_k^\top \mathbf{1}} \quad \text{and} \quad \mathbf{P}_k = \hat{\mathbf{P}}_k \operatorname{diag}(\frac{\mathbf{a}_{k+1}}{\hat{\mathbf{P}}_k^\top \mathbf{1}})$$

Similarly, denote $\mathbf{P}_k$ as the projection on $\mathcal{C}_{2k}$ of $\overline{\mathbf{P}}_k$. The first-order conditions of $\operatorname{Proj}_{\mathcal{C}_{2k}}^{KL}(\overline{\mathbf{P}}_k)$ states the existence of Lagrange multipliers $\mathbf{f}_k$ such that:

$$\varepsilon \log\frac{\mathbf{P}_k}{\overline{\mathbf{P}}_k} + \mathbf{f}_k\mathbf{1} = 0$$

Denote $\mathbf{u}_k = e^{-\mathbf{f}_k/\varepsilon}$. Condition $\mathbf{P_k}\mathbf{1} = \mathbf{a}_k$ thus implies that

$$\mathbf{u}_k = \frac{\mathbf{a}_k}{\overline{\mathbf{P}}_k\mathbf{1}} \quad \text{and} \quad \mathbf{P}_k = \operatorname{diag}(\frac{\mathbf{a}_k}{\overline{\mathbf{P}}_k\mathbf{1}})\overline{\mathbf{P}}_k$$

Finally, by leveraging Lagrange multiplier function on $\mathbf{a}_k$, we get $\forall k = 1, ..., K - 1$:

$$f_k + g_{k-1} = 0$$

which implies $u_k \odot v_{k-1} = \mathbf{1}$, and thus we get the desired equation for $\mathbf{a}_k$:

$$\left( \frac{\mathbf{a}_k}{\mathbf{P}_k \mathbf{1}} \right) \odot \left( \frac{\mathbf{a}_k}{\mathbf{P}_{k-1}^\top \mathbf{1}} \right) = \mathbf{1} \quad \Rightarrow \quad \mathbf{a}_k = (\mathbf{P}_k \mathbf{1})^{1/2} \odot \left( \mathbf{P}_{k-1}^\top \mathbf{1} \right)^{1/2}$$

## F    PROOF OF REGULARIZED MLOT-SINKHORN SOLUTION AND ITERATION FORM

**The case $\tau = 0$.**

The entropic regularized MLOT can be formulated as

$$\min_{\{\mathbf{P}_k\},\{\mathbf{a}_k\}} \sum_{k=1}^{K-1} \left( \langle \mathbf{C}_k, \mathbf{P}_k \rangle - \epsilon H(\mathbf{P}_k) \right) - \tau \sum_{k=2}^{K-1} H(\mathbf{a}_k) \tag{26}$$

subject to

$$\mathbf{P}_k \mathbf{1} = \mathbf{a}_k \quad \text{and} \quad \mathbf{P}_k^\top \mathbf{1} = \mathbf{a}_{k+1} \quad \forall k = 1, \dots, K - 1. \tag{27}$$

The Lagrange multiplier function is

$$L = \sum_{k=1}^{K-1} \left( \langle \mathbf{C}_k, \mathbf{P}_k \rangle - \epsilon H(\mathbf{P}_k) \right) - \tau \sum_{k=2}^{K-1} H(\mathbf{a}_k)$$
$$- \sum_{k=1}^{K-1} \langle \mathbf{f}_k, \mathbf{P}_k \mathbf{1} - \mathbf{a}_k \rangle - \langle \mathbf{g}_k, \mathbf{P}_k^\top \mathbf{1} - \mathbf{a}_{k+1} \rangle \tag{28}$$

Firstly,

$$\frac{\partial L}{\partial \mathbf{P}_k} = \mathbf{C}_k + \varepsilon \log \mathbf{P}_k - \mathbf{f}_k \mathbf{1}^\top - \mathbf{1}^\top \mathbf{g}_k = 0$$
$$\Rightarrow \mathbf{P}_k = \text{diag}\left( e^{\mathbf{f}_k/\varepsilon} \right) \cdot e^{-\mathbf{C}_k/\varepsilon} \cdot \text{diag}\left( e^{\mathbf{g}_k/\varepsilon} \right) \tag{29}$$

Set that: $\mathbf{u}_k = e^{\mathbf{f}_k/\varepsilon}, \mathbf{v}_k = e^{\mathbf{g}_k/\varepsilon}, \mathbf{S}_k = e^{-\mathbf{C}_k/\varepsilon}$, we have:

$$\mathbf{P}_k = \text{diag}(\mathbf{u}_k) \mathbf{S}_k \text{diag}(\mathbf{v}_k) \tag{30}$$

Due to $\mathbf{P}_k \mathbf{1} = \mathbf{a}_k$ and $\mathbf{P}_k^\top \mathbf{1} = \mathbf{a}_{k+1}$ We have:

$$\mathbf{u}_k = \frac{\mathbf{a}_k}{\mathbf{S}_k \mathbf{v}_k}, \quad \mathbf{v}_k = \frac{\mathbf{a}_{k+1}}{\mathbf{S}_k^\top \mathbf{u}_k} \tag{31}$$

What's more, when $\tau = 0$:

$$\frac{\partial L}{\partial \mathbf{a}_k} = \mathbf{f}_k + \mathbf{g}_{k-1} = 0 \tag{32}$$

Thus, $\mathbf{u}_k \odot \mathbf{v}_{k-1} = \mathbf{1}$ Then we have:

$$\frac{\mathbf{a}_k}{\mathbf{S}_k \mathbf{v}_k} \odot \frac{\mathbf{a}_k}{\mathbf{S}_{k-1}^\top \mathbf{u}_{k-1}} = 1$$
$$\mathbf{a}_k = \left[ (\mathbf{S}_k \mathbf{v}_k) \odot (\mathbf{S}_{k-1}^\top \mathbf{u}_{k-1}) \right]^{\frac{1}{2}}, \quad \text{for } k = 2, ..., K - 1 \tag{33}$$

**The case $\tau > 0$.**

The Lagrange multiplier function is

$$L = \sum_{k=1}^{K-1} \left( < \mathbf{C}_k, \mathbf{P}_k > - \epsilon H(\mathbf{P}_k) \right) - \tau \sum_{k=2}^{K-1} H(\mathbf{a}_k)$$
$$- \sum_{k=1}^{K-1} < \mathbf{f}_k, \mathbf{P}_k \mathbf{1} - \mathbf{a}_k > - < \mathbf{g}_k, \mathbf{P}_k^\top \mathbf{1} - \mathbf{a}_{k+1} > \tag{34}$$

Firstly,

$$\frac{\partial L}{\partial \mathbf{P}_k} = \mathbf{C}_k + \varepsilon \log \mathbf{P}_k - \mathbf{f}_k \mathbf{1}^\top - \mathbf{1}^\top \mathbf{g}_k = 0$$

$$\Rightarrow \mathbf{P}_k = \operatorname{diag}\left(e^{\mathbf{f}_k/\varepsilon}\right) \cdot e^{-\mathbf{C}_k/\varepsilon} \cdot \operatorname{diag}\left(e^{\mathbf{g}_k/\varepsilon}\right)$$

(35)

Set that: $\mathbf{u}_k = e^{\mathbf{f}_k/\varepsilon}, \mathbf{v}_k = e^{\mathbf{g}_k/\varepsilon}, \mathbf{S}_k = e^{-\mathbf{C}_k/\varepsilon}$, we have:

$$\mathbf{P}_k = \operatorname{diag}(\mathbf{u}_k)\mathbf{S}_k \operatorname{diag}(\mathbf{v}_k)$$

(36)

Due to $\mathbf{P}_k \mathbf{1} = \mathbf{a}_k$ and $\mathbf{P}_k^\top \mathbf{1} = \mathbf{a}_{k+1}$ We have:

$$\mathbf{u}_k = \frac{\mathbf{a}_k}{\mathbf{S}_k \mathbf{v}_k}, \quad \mathbf{u}_k = \frac{\mathbf{a}_{k+1}}{\mathbf{S}_k^\top \mathbf{u}_k}$$

(37)

What's more, when $\tau > 0$

$$\frac{\partial L}{\partial \mathbf{a}_k} = \tau \log \mathbf{a}_k + \mathbf{f}_k + \mathbf{g}_{k-1} = 0$$

$$\mathbf{a}_k = (\mathbf{u}_k \odot \mathbf{v}_{k-1})^{-\epsilon/\tau}$$

(38)

# G   PROOF OF EQUIVALENCE BETWEEN MLOT AND ITS KL-DIVERGENCE FORM

From the definition of $\widetilde{KL}$ and $(\mathbf{S}_k)_{ij} = e^{-(\mathbf{C}_k)_{ij}/\epsilon}$, we have

$$\sum_{k=1}^{K-1} \widetilde{KL}(\mathbf{P}_k|\mathbf{S}_k) = \sum_{k=1}^{K-1} \sum_{ij} \left( (\mathbf{P}_k)_{ij} \log(\mathbf{P}_k)_{ij} - (\mathbf{P}_k)_{ij} + (\mathbf{P}_k)_{ij}\frac{(\mathbf{C}_k)_{ij}}{\varepsilon} + (\mathbf{S}_k)_{ij} \right)$$

$$= \sum_{k=1}^{K-1} \sum_{ij} \left( (\mathbf{P}_k)_{ij} \left(\log(\mathbf{P}_k)_{ij} - 1\right) + \frac{1}{\epsilon}(\mathbf{P}_k)_{ij}(\mathbf{C}_k)_{ij} + (\mathbf{S}_k)_{ij} \right) \quad (39)$$

$$= \frac{1}{\epsilon} \sum_{k=1}^{K-1} \langle \mathbf{C}_k, \mathbf{P}_k \rangle - \varepsilon H(\mathbf{P}_k) + \text{Const}.$$

and

$$\sum_{k=2}^{K-1} \widetilde{KL}(\mathbf{a}_k|\mathbf{\Delta}_k) = \sum_{k=2}^{K-1} \sum_i (\mathbf{a}_k)_i \left(\log(\mathbf{a}_k)_i + \log n_k - 1\right)$$

$$= \sum_{k=2}^{K-1} \sum_i (\mathbf{a}_k)_i \left(\log(\mathbf{a}_k)_i - 1\right) + \log n_k \sum_i (a_k)_i \quad (40)$$

$$= \frac{1}{\tau} \sum_{k=2}^{K-1} H(\mathbf{a}_k) + \text{Const}.$$

Notice that the Const in expression is irrelevant when it comes to solving optimization problems. Therefore $\min_{(\mathbf{P})_k, (\mathbf{a})_k} \varepsilon \sum_{k=1}^{K-1} \widetilde{KL}(\mathbf{P}_k|\mathbf{S}_k) + \tau \sum_{k=2}^{K-1} \widetilde{KL}(\mathbf{a}_k|\mathbf{\Delta}_k)$ is exactly equivalent to Eq. 3.

# H   PROOF OF MLOT CONVERGENCE WITH $\varepsilon$ AND $\tau$

**Convergence with $\varepsilon$**   In this part, we prove that the entropic regularization on couplings will converge to original MLOT. We consider a sequence $(\varepsilon_l)_l > 0$ such that $\varepsilon_l \to 0$. We denote $(\mathbf{P}_k^{\varepsilon_l})_k$ as the optimal solution of Eq. 3 with $\varepsilon = \varepsilon_l, \tau = 0$, and denote $(\mathbf{P}_k^\star)_k$ as the optimal solution of Eq. 2. By optimality of $(\mathbf{P}_k^{\varepsilon_l})_k$ and $(\mathbf{P}_k^\star)_k$ for their respective optimization problems, we have:

$$\sum_{k=1}^{K-1} \langle \mathbf{C}_k, \mathbf{P}_k^{\varepsilon_l} \rangle - \varepsilon_l H(\mathbf{P}_k^{\varepsilon_l}) \;\leqslant\; \sum_{k=1}^{K-1} \langle \mathbf{C}_k, \mathbf{P}_k^\star \rangle - \varepsilon_l H(\mathbf{P}_k\star)$$

$$\sum_{k=1}^{K-1} \langle \mathbf{C}_k, \mathbf{P}_k^\star \rangle \;\leqslant\; \sum_{k=1}^{K-1} \langle \mathbf{C}_k, \mathbf{P}_k^{\varepsilon_l} \rangle$$

(41)

Therefore:

$$0 \leqslant \sum_{k=1}^{K-1} \langle \mathbf{C}_k, \mathbf{P}_k^{\varepsilon_l} - \mathbf{P}_k^{\star} \rangle \quad \leqslant \quad \sum_{k=1}^{K-1} \varepsilon_l \left[ H(\mathbf{P}_k^{\varepsilon_l}) - H(\mathbf{P}_k^{\star}) \right] \tag{42}$$

Since entropic function $H(\mathbf{P})$ is continuous and inner product here is always positive, the limitation $\varepsilon_l \to 0$ shows that $\mathbf{P}_k^{\varepsilon_l} = \mathbf{P}_k^{\star}$, $\forall k = 1, 2, ..., K-1$, which proves Eq. 4.

**Convergence with $\tau$**    In this part, we prove that the entropic regularization on both couplings and intermediates will converge to the problem that only regularize couplings, given the fixed $\varepsilon_0$. We consider a sequence $(\tau_l)_l > 0$ such that $\tau_l \to 0$. We denote $(\mathbf{P}_k^{\tau_l})_k$ as the optimal solution of Eq. 3 with $\varepsilon = \varepsilon_0, \tau = \tau_l$, and denote $(\mathbf{P}_k^{\varepsilon_0})_k$ as the optimal solution of Eq. 3 without regularization on intermediates. By optimality of $(\mathbf{P}_k^{\tau_l})_k$ and $(\mathbf{P}_k^{\varepsilon_0})_k$ for their respective optimization problems, we have:

$$\sum_{k=1}^{K-1} \langle \mathbf{C}_k, \mathbf{P}_k^{\tau_l} \rangle - \varepsilon_0 H(\mathbf{P}_k^{\tau_l}) - \tau_l \sum_{k=2}^{K-1} H(\mathbf{a}_k^{\tau_l}) \quad \leqslant \quad \sum_{k=1}^{K-1} \langle \mathbf{C}_k, \mathbf{P}_k^{\varepsilon_0} \rangle - \varepsilon_0 H(\mathbf{P}_k^{\varepsilon_0}) - \tau_l \sum_{k=2}^{K-1} H(\mathbf{a}_k^{\varepsilon_0})$$

$$\sum_{k=1}^{K-1} \langle \mathbf{C}_k, \mathbf{P}_k^{\varepsilon_0} \rangle - \varepsilon_0 H(\mathbf{P}_k^{\varepsilon_0}) \quad \leqslant \quad \sum_{k=1}^{K-1} \langle \mathbf{C}_k, \mathbf{P}_k^{\tau_l} \rangle - \varepsilon_0 H(\mathbf{P}_k^{\tau_l})$$

$$\tag{43}$$

Therefore:

$$0 \leqslant \sum_{k=1}^{K-1} \langle \mathbf{C}_k, \mathbf{P}_k^{\tau_l} - \mathbf{P}_k^{\varepsilon_0} \rangle - \varepsilon_0 \left[ H(\mathbf{P}_k^{\tau_l}) - H(\mathbf{P}_k^{\varepsilon_0}) \right] \quad \leqslant \quad \sum_{k=2}^{K-1} \tau_l \left[ H(\mathbf{a}_k^{\tau_l}) - H(\mathbf{a}_k^{\varepsilon_0}) \right] \tag{44}$$

Similarly, since entropic function $H(\mathbf{a})$ is continuous, the limitation $\tau_l \to 0$ shows that regularization on intermediate can converge to non-regularization on intermediate:

$$\sum_{k=1}^{K-1} \langle \mathbf{C}_k, \mathbf{P}_k^{\tau_l} \rangle - \varepsilon_0 H(\mathbf{a}_k^{\tau_l}) = \sum_{k=1}^{K-1} \langle \mathbf{C}_k, \mathbf{P}_k^{\varepsilon_0} \rangle - H(\mathbf{a}_k^{\varepsilon_0}).$$

## I    CONVERGENCE OF MLOT RESPECTED TO $\tau$

As mentioned in Section 4, Fig 10 and Fig 11 visualize the convergence of MLOT-Sinkhorn with respect to $\varepsilon$ and $\tau$.

The shade of color in the heatmaps indicates the magnitude of the transport values at each location, while the central bar graphs represent the intermediate distributions computed by the algorithm. This experiment aims to showcase the convergence properties regarding $\varepsilon$ and $\tau$ as proven in Prop. 1.

The experiment is conducted on Line dateset, with $N = 100$, $K = 3$, $(n_k)_k = \{25, 50, 25\}$, $D = 5$, where points in each layer are uniformly distributed along a line of length 20. Both the source and target distributions were randomly generated and normalized.

In Fig. 10, $\tau$ is set to 0, and a series of decreasing $\varepsilon$ values are employed, comparing to the ground truth solution of Eq. 2 ($\varepsilon = 0$), showing the convergence of MLOT-Sinkhorn with respect to $\varepsilon$.

In Fig. 11, $\varepsilon$ is fixed as $1 \times 10^{-3}$, and a series of decreasing $\tau$ values are employed, showing the convergence of MLOT-Sinkhorn with respect to $\tau$.

## J    ARCHIMEDEAN DISTANCE BETWEEN TWO POINTS

Archimedes' spiral is curve expressed as $r(\theta) = b(\theta - \theta_0)$. Suppose two a spiral passes through two points $(r_1, \theta_1)$, $(r_2, \theta_2)$. The curve's parameters can be determined as:

$$b = \frac{r_2 - r_1}{\theta_2 - \theta_1}, \quad \theta_0 = \frac{\theta_1 r_2 - \theta_2 r_1}{r_2 - r_1} \tag{45}$$

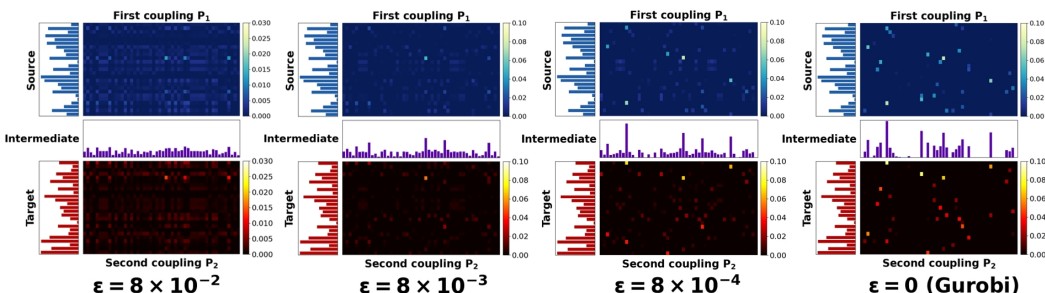

Figure 10: Impact of $\varepsilon$ on the MLOT-Sinkhorn algorithm solutions, generated by varying $\varepsilon = 8 \times 10^{-2}, 8 \times 10^{-3}, 8 \times 10^{-4}$, and 0 (Gurobi) with $\tau = 0$, on Line data. As $\varepsilon$ decreases, the solution of our algorithm converges towards the exact solution of Eq. 2.

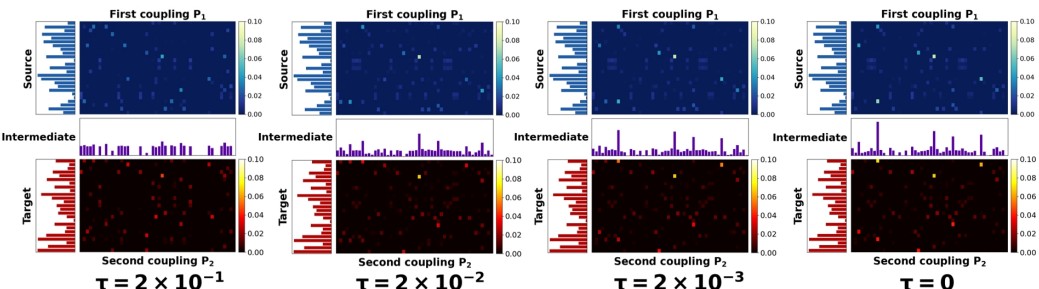

Figure 11: Impact visualization of $\tau$ on the MLOT-Sinkhorn. The experiment is conducted on Line data, by fixed $\varepsilon = 1 \times 10^{-3}$ and varying $\tau = 2 \times 10^{-1}, 2 \times 10^{-2}, 2 \times 10^{-3}$, and 0 (without regularization on intermediate). As $\tau$ decreases, the solution progressively converges towards the solution without regularization on intermediate.

The length of the curve is:

$$
\begin{aligned}
\mathrm{d}l &= \sqrt{\mathrm{d}r^2 + (r\mathrm{d}\theta)^2} \\
\Rightarrow \quad L &= \int_{r_1}^{r_2} \sqrt{1 + \frac{r^2}{b^2}}\,\mathrm{d}r \\
&= \frac{r}{2b}\sqrt{b^2 + r^2} + \frac{b}{2}\ln\left(r + \sqrt{b^2 + r^2}\right)\Bigg|_{r_1}^{r_2}
\end{aligned}
\tag{46}
$$

Under the circumstances in Ring Data, where the radii of neighbouring rings differ by 1, thus $b = 1/(\theta_2 - \theta_1)$. Further denote $\theta_2 - \theta_1$ as $a$. Let:

$$
F(r) = \frac{r}{2}\sqrt{1 + a^2 r^2} + \frac{1}{2a}\ln\left(ar + \sqrt{1 + a^2 r^2}\right)
\tag{47}
$$

Then the Archimedean distance between two points can be written as $F(r_2) - F(r_1)$.

## K  RELATION TO THE DYNAMIC OT AND SCHRÖDINGER BRIDGE

Fundamentally, our MLOT is akin to Dynamic Optimal Transport [35] in that both can be seen as calculating the intermediate steps of the entire transport process. The difference lies in the fact that we fix the positions of each layer or the cost matrices between two layers in our MLOT, while in Dynamic OT, the locations are continuous throughout the entire space. The relationship between the Schrödinger bridge [10] and our entropic MLOT is similar to the relationship between the aforementioned two OT variants; both can be regarded as special cases in a discrete state. Therefore, our MLOT can offer new perspectives and approximate computations for Dynamic OT and the Schrödinger bridge.

| | SBP | LOT | MLOT |
|---|---|---|---|
| **Premise** | Continuous flow in space | Factorized by low-rank middle anchors | Sequential flow through fixed multi-stage layers |
| **Intermediate State** | Probability distributions | Supports' coordinate | Mass distribution |
| **Optimization Variable** | Probability distributions $p_t(x)$ over $t \in (0,1)$ | Anchors' position $z_j$ and transportation | Transportation series $\{P_k\}$ |
| **Cost** | Entropic regularized OT cost | k-Wasserstein barycenter | (sum of) Primal OT cost |
| **Algorithm** | Iterative Proportional Fitting | Lloyd-type | Mirror Descent |

Table 4: Comparison of SBP, LOT, and MLOT.

## L  CONVEXITY OF MLOT

We show that MLOT formulation Eq. 2 is a convex optimization problem (also linear programming).

Firstly, the inner-product and summation in objective function is linear.

Secondly, we show that constraints part is linear. Let:

$$\mathbf{A}_k = \left[ \begin{array}{c} \mathbf{1}_{n_k}^\top \otimes \mathbf{I}_{n_{k+1}} \\ \mathbf{I}_{n_k} \otimes \mathbf{1}_{n_{k+1}}^\top \end{array} \right] \in \mathbb{R}^{(n_k + n_{k+1}) \times n_k n_{k+1}}$$

where $\otimes$ is Kronecker's product, $\mathbf{I}_n$ is identity matrix by $n$ size. Intuitively, this is for computing the row-sum and col-sum of a vectorized matrix.

Then the constraints can be re-formulate to linear form:

$$\mathbf{A}_k \cdot \text{vec}(\mathbf{P}_k) = \left[ \begin{array}{c} \mathbf{a}_{k-1} \\ \mathbf{a}_k \end{array} \right] \quad , \forall k = 1, .., K-1$$

where denote $\mathbf{a}_0 = s$, $\mathbf{a}_{K-1} = t$ be the known fixed distribution, and other $\mathbf{a}_k \in \Delta_{n_k}$ is restrained in $n_k$-dim simplex, which is also a linear constraint.

Therefore MLOT problem (Eq. 2) is LP, thus also convex problem.

## M  OVERALL TIME COMPLEXITY

To prove the overall complexity of MLOT-Sinkhorn Alg. 1, we refer to the technique used in [2] to adapt to our algorithm.

The time spent can be decomposed into two part: "Complexity per Iteration" × "Iteration number before convergence/stop".

The first part is easy to analysis, since each Sinkhorn-based algorithm is simply matrix-scale type method. Each iteration cost $O((K-1)n^2)$.

Since MLOT-Sinkhorn do the update for each layer respectively, we make an important assumption that, we regard its stop criteria's property follows summation of a series classic Sinkhorn.

Following [2], let $s_k = \sum_{ij} \exp(-\eta C_k), l_k = \min_{ij} \exp(-\eta C_k)$. Thus, to get a $\epsilon'$-error result, MLOT-Sinkhorn needs $\mathcal{O}(\epsilon'^{-2} \cdot \sum \log(s_k/l_k))$ iterations.

To make the result adaptive with more convenient parameters, we use the following scaling inequality to substitute $s_k, l_k$:

$$\log(s_k/l_k) = \log(s) + \log(1/l) \leq \mathcal{O}(\log n + \eta \|C_k\|_\infty)$$

Let $\eta = \dfrac{4(K-1)\log(n)}{\epsilon}$, $\epsilon' = \dfrac{\epsilon}{8(K-1)L}$, where $L = \max_k \|C_k\|_\infty = \max_k \max_{ij}(C_k)_{ij}$, we get the Iteration needs before getting a $\epsilon$-error solution is: $\mathcal{O}((K-1)^4 L^3 \epsilon^{-3}\log(n))$. Thus overall complexity is:

$$\mathcal{O}\left((K-1)^5 L^3 \epsilon^{-3} n^2 \log(n)\right)$$

