# OpenReview forum: "MLOT: Generalizing the Bipartite Structure to a Multi-Layered Framework for Optimal Transport"
_ICLR.cc/2026/Conference — Submitted to ICLR 2026_

### Official Review · Reviewer_gBDq · 2025-10-26

**Soundness:** 2
**Presentation:** 2
**Contribution:** 2
**Rating:** 4
**Confidence:** 3

**Summary:**

This paper proposes a multi-layer optimal transport (MLOT) method that extends the original optimal transport formulation. The theoretical properties of MLOT are discussed. Two theoretically guaranteed algorithms are designed to solve MLOT. The authors further conduct experiments on text-image retrieval and visual graph matching.

**Strengths:**

The MLOT generalizes both OT and low-rank OT and looks interesting.

**Weaknesses:**

1. Line 190 "The optimization described above is essentially a convex optimization problem" is not very obvious. The authors may consider formalizing this using a proposition and provide formal proofs.

2. Proposition 1 looks not very clean. For example, can $\epsilon_0$ be 0?

3. In Line 211, why is the Bregman iterative algorithm a Sinkhorn-based algorithm?

4. It would be better to directly provide a computational complexity at Line 239 - Line 241.

5. For Line 314 - Line 317, what's the advantage of using augmented data? Are there any disadvantages of using augmented data?

6. In Figure 6, the generated interpolations images are not of high quality. Many of them look like simple weighted sum of the source image and the target image at the pixel level.

7. Equation (34) seems wrong.

8. It is still not very clear why MLOT is more adaptable to complex structures found in deep learning tasks.

**Questions:**

See above.

---

> ### Author Response · Authors · 2025-11-20
> **Response to reviewer (Part 1)**
>
> **We sincerely thanks for your detailed and constructive questions**. Hope the following explanation addresses your concerns.
>
> > Q1: "MLOT is a convex optimization problem" is not very obvious.
>
> ***Since rebuttal cannot render {array}, the following matrix notation may be confused.***
>
> ***We have supplemented this part in Appendix.L.1***
>
> Thanks for pointing this out. **Eq.(2) is convex**, and more specifically, **a linear programming.**
>
> [1/2] The inner-product and summation in objective function is linear.
>
> [2/2] For constraints part, let: $A_k=[1_{n_k}^{\top} \otimes I_{n_{k+1}}$ ; $I_{n_k} \otimes 1_{n_{k+1}}^{\top}] \in \mathbb{R}^{\left(n_k+n_{k+1}\right) \times n_k n_{k+1}}$
>
> where $\otimes$ is Kronecker’s product. (Intuitively, this is for computing the row-sum and col-sum of a vectorized matrix.)
>
> Then the constraints can be re-formulate to linear form: $A_k \cdot\text{vec}(P_k) = [a_{k-1}$ ; $a_{k}] \quad,\forall k=1,..,K-1$
>
> where denote $a_0=s,a_{K-1}=t$. And other $a_k\in\Delta_{n_k}$ is restrained in $n_k$-dim simplex, which is also a linear constraint.
>
> Therefore MLOT problem (Eq.2) is LP, thus also convex problem.
>
> > Q2: Prop.1 Can $\epsilon_0$ be 0?
>
> No. The convergence for $\tau$ is on any given $\epsilon=\epsilon_0>0$.
>
> Eq.4+Eq.5 is sufficient to prove the joint limit $P^{(\epsilon,\tau)}\to P^{(0,0)}$ for all $\epsilon,\tau>0$ tends to 0 together in any order.
>
> > Q3: Why is the Bregman iterative algorithm a Sinkhorn-based algorithm?
>
> [1/3] More precisely, these two algorithms are both **matrix-scaling-based method**. Sinkhorn algorithm is firstly invented by [2] in 1966, and is simply a matrix scaling algorithm, which do nothing with OT.
>
> [2/3] After Cuturi propose entropic regularization on OT [3] in 2013, people find that Sinkhorn exactly solve this problem.
>
> [3/3] In 2015, Cuturi [4] proved that Sinkhorn is equivalent to KL-projection (alternating Bregman Projection in KL geometry).
>
> > Q4: It would be better to directly provide a computational complexity.
>
> ***If the following formula is not rendered properly, we have added this part in Appendix.L.2***
>
> It's a little bit tricky to give overall time complexity. We refer to the technique used in [1] to adapt to our algorithm.
>
> The time spent can be decomposed into "Complexity per Iter $\times$ Iter number before stop"
>
> Let $s_k=\sum_{ij} \exp{(-\eta C_k)}, l_k=\min_{ij}\exp{(-\eta C_k)}$. Since MLOT-Sinkhorn do the update for each layer respectively, we regard its property follows summation of a series classic Sinkhorn. Thus, to get a $\epsilon'$-error result, MLOT-Sinkhorn needs $O(\epsilon'^{-2}\cdot \sum{\log(s_k/l_k)})$.
>
> (For the purpose of substitute $s_k,l_k$) Let $\eta=\frac{4(K-1)\log(n)}{\epsilon},~\epsilon'=\frac{\epsilon}{8(K-1)L}$, where $L=\max_{k}\max_{ij} (C_{k})_{ij}$, we get "Iter number before stop" is: $O((K-1)^4L^3\epsilon^{-3}\log(n))$
>
> Each iteration cost $O((K-1)n^2)$. Thus overall complexity to get a $\epsilon$-error solution is: $O((K-1)^5 L^3\epsilon^{-3} n^2\log(n))$
>
> > Q5: What's the advantage of using augmented data? Are there disadvantages?
>
> [1/3] Data augmentation is already widely adopted in Contrastive Learning(e.g. CLIP), to guide the model to **focus on invariances and essential patterns**.
>
> [2/3] The paper OT-CLIP[2] interprets CL through **point set matching**, and **formulat it as a (inverse)OT problem** between two set. [2] further prove that **Softmax is the optimal solution of this special EntropicOT problem.**
>
> [3/3] What's new in our paper is that, extending the line of OT-CLIP[2], we propose a framework that **contrast mulitiple pos/neg samples at same time**. To achieve this, we **formulate this as MLOT problem**. Vanilla Softmax cannot solve this, since it's bi-layer. **MLOT functions as a multi-layered Softmax** in our application.
>
> > Q6: Fig.6 interpolations images are not of high quality.
>
> [1/2] We agree that, for **high-resolution color image** datasets, the interpolations produced by MLOT are not substantially different from barycenter-based approaches. And, as you mentioned, are not of high quality.
>
> [2/2] However, the more important thing we want to emphasize is that, MLOT has the **structure very similar to this interpolation computation task.** Each layer represents one middle image, and MLOT outputs **whole image path in a single run.** In contrast, barycenter based method needs **vary weights to generate each interpolation.**
>
> > Q7: Eq.(34) seems wrong.
>
> Thank you for checking our proof detailedly. We have fixed this typo in paper.
>
> ---
> [1] Jason Altschuler, Jonathan Weed, and Philippe Rigollet. Near-linear time approximation algorithms for optimal transport via Sinkhorn iteration. Arxiv, 2017.
>
> [2] A Relationship between Arbitrary Positive Matrices and Stochastic Matrices.
>
> [3] Sinkhorn Distances: Lightspeed Computation of Optimal Transportation Distances
>
> [4] Iterative Bregman Projections for Regularized Transportation Problems

---

> ### Author Response · Authors · 2025-11-20
> **Response to reviewer (Part 2)**
>
> Thanks for giving us opportunity to further explain. This thoughtful question is quite important and should be clarified.
>
> > Q8: Why MLOT is more adaptable to complex structures found in DL tasks.
>
> [1/3] **Entropic MLOT can be viewed as multi-layered Softmax**, thus adaptable to Contrastive Learning with multiple sample sets. Let us explain in detail.
>
> [2/3] OT-CLIP[2] gives an insight on relationship among Contrastive Learning, Softmax and OT. It shows that CL can be viewed as **point-set matching**, and thus be **formulated as (inverse)OT** problem. It further proves **Softmax is the optimal solution of this special Entropic OT.**
>
> [3/3] **Extending the line of OT-CLIP**, which only deal with two point-set, **we proposed a framework that contrast multiple pos/neg samples (generated by data augmentation)**. To solve this, we **formulate it as a MLOT** problem.
>
> We refer the "complex structure" to the multiple contrastive samples/distributions. Besides the application in (self)supervised representation learning already shown in paper, other tasks also imply multiple data distribution sets, such as Mix-up Training.
>
> ---
> [2] Understanding and Generalizing Contrastive Learning from the Inverse Optimal Transport Perspective. ICML.2023

---

> > ### Comment · Reviewer_gBDq · 2025-11-27
> > **Thank the authors for the great response**
> >
> > Thank the authors for the great response! Most of my concerns have been addressed. Below is one minor additional comment.
> >
> > Q5: What's the advantage of using augmented data? Are there disadvantages?
> >
> > The authors seem to have missed the second half of the question. Do the authors have any insights or experience regarding the circumstances where data augmentation does not work.

---

> ### Author Response · Authors · 2025-11-27
> **Response to reviewer**
>
> We're glad that our responses have addressed most of your concerns.
>
> [1/3] A potential limitation of introducing augmented data comes from numerical OT problems, which are highly sensitive to exact numerical values.
>
> [2/3] In our paper, experiments involving augmented data are conducted in **image / text modalities**, where small **local perturbations typically do not alter the overall semantic meaning.** Thus augmented data can function effectively.
>
> [3/3] In **accuracy-oriented** numerical OT tasks, such as AI4PDE, AI4CO, unlike in **perceptual domains**, augmentations that are **not strictly value-preserving** may lead to degraded performance.
>
> ---
> Given that your concerns have now been clarified, **we kindly ask you to consider revisiting your rating and confidence score. we remain fully committed to addressing any additional questions you may have.**

---

### Official Review · Reviewer_Fbf2 · 2025-10-31

**Soundness:** 3
**Presentation:** 3
**Contribution:** 2
**Rating:** 4
**Confidence:** 4

**Summary:**

This paper introduces Multi-Layered Optimal Transport (MLOT), a novel framework that generalizes traditional optimal transport from a bipartite structure to a multi-layer graph. The authors develop an entropically regularized formulation and derive efficient Sinkhorn-style algorithms.

**Strengths:**

1) The core idea of generalizing OT from a bipartite to a multi-layered graph is both interesting and important. It elegantly addresses a key limitation of vanilla OT in deep learning, where data often has a more complex, hierarchical, or dynamic structure.

2) MLOT-Sinkhorn is shown to be highly efficient and accurate on synthetic data, scaling to problems where baseline solvers like Gurobi fail.

3) Theoretical justification of the  global convergence.

**Weaknesses:**

1) The authors correctly identify the conceptual proximity of their multi-layered framework to Dynamic Optimal Transport and the Schrödinger Bridge Problem (SBP), viewing MLOT as a discrete-state analogue. However, this connection remains under-explored.

2) The propagation chain on MLOT-Sinkhorn algorithm creates feedback loops where the output of one layer's update is immediately used as input for another's within the same iteration. The proof of Proposition 4 in Appendix D may be incorrect, because it fails to account for this temporal and spatial coupling, treating the layers as if they could be analyzed in isolation and then combined with a simple worst-case bound. The updates are not independent across layers, creating a propagation chain that's not captured:

$u_k^{l+1} = a_k^l * (S_k v_k^l)$

$v_k^{l+1} = a_{k+1}^l * (S_k^⊤ u_k^{l+1})  $

$a_k^{l+1} = ((S_{k-1}^⊤ u_{k-1}^{l+1}) * (S_k v_k^{l+1}))^{1/2}$

3) The stated limitation ("our algorithm cannot obtain the exact solution or deal with more general graphs") is too brief and can be expanded.

**Questions:**

Given the stated connection to Generalized Schrödinger Bridges (GSB), is it possible to provide a quantitative comparison against specialized GSB solvers (for example HOTA [1]) on a standard 2D benchmark (Stunnel, Vneck, and GMM ref. [1])?

[1] Buzun et. al.  HOTA: Hamiltonian framework for Optimal Transport Advection. https://arxiv.org/abs/2507.17513

---

> ### Author Response · Authors · 2025-11-26
> **Response to related work & limitations**
>
> **We apologize for the delayed response. We took your comments very seriously, since your questions are insightful and we have been actively working on addressing them thoroughly.**
>
> > W1: More detailed explanation on connection between SBP and MLOT.
>
> We'd like to further explain the relationship between SBP, MLOT and another similar structured Low-Rank OT (as discussed in paper).
>
> [1/4] Beyond the apparent distinction between continuous and discrete scenario, their fundamental differences lie in their **modeling focus and optimization objectives.**
>
> [2/4] SBP models transportation **flow**, where the **distribution and supports along the path** are optimized with respect to the reference measure $P_{ref}$.
>
> [3/4] LOT is conceptually closer to SBP: it **factorizes** OT into 2 phases via a layer of **unknown anchors** (see Fig.1.c), and the core optimization is on these **intermediates' support**.
>
> [4/4] MLOT also **factorizes** OT into multiple steps, but the **supports at each intermediate layer are fixed**. The core optimization is on **distribution of intermediates.**
>
> The essential relationship is listed as following, and we have supplemented this in Appendix.K.
>
> |                           | SBP                            | LOT                                        | MLOT                                             |
> | ------------------------- | ------------------------------ | ------------------------------------------ | ------------------------------------------------ |
> | **Premise**               | Continuous flow in space       | Factorized by low-rank middle anchors      | Sequential flow through fixed multi-stage layers |
> | **Intermediate State**    | Probability distributions      | Supports' coordinate                       | Mass distribution                                |
> | **Optimization Variable** | $p_t(x)$ over $t\in(0,1)$      | Anchors' position $z_j$ and transportation | Transportation series $P_k$                      |
> | **Cost**                  | Entropic regularized OT cost   | k-Wasserstein barycenter                   | (sum of) Primal OT cost                          |
> | **Algorithm**             | Iterative Proportional Fitting | Lloyd-type                                 | Mirror Descent                                   |
>
>
> > W3: Limitation part is too brief.
>
> [1/3] As discussed in related work of Graph-OT, MLOT can be seen as a Graph-OT problem on a **special graph**, which has layered structure, without cross-layer edges. Under this condition, MLOT provides a **shortcut** to compute optimal transport flow **without computing shortest-path on graph**.
>
> [2/3] Our algorithm's benefit is realized only in such structure currently. MLOT is limited for general graph setting.
>
> [3/3] We are considering to adaptively learning the layer's support (i.e. auto-layering) for genral graph in the future work.
>
> > Q1: Is it possible to provide a quantitative comparison against specialized GSB solvers (e.g. HOTA) on a standard 2D benchmark?
>
> As detailed discussed in W1, we here make a summary to highlight the distinction: SBP and LOT aim to optimize intermediate **supports and distributions**, while MLOT fixes supports and optimizes the **multi-step transport flow**.
>
> Thus ***2D benchmark for SBP is not compatible for our MLOT.*** Actually it's **more aligned with LOT scenario.** (Though current algorithms in Low-rank OT and Factor OT[1][2] only deal with 1-layer middle anchor, which is not enough to handle multiple time steps in SBP.)
>
> ---
> [1] Statistical Optimal Transport via Factored Couplings. AISTATS.2019.
>
> [2] Making transport more robust and interpretable by moving data through a small number of anchor points. ICML.2021

---

> ### Author Response · Authors · 2025-11-26
> **Response to issue regarding convergence proof in Appendix.D**
>
> > W2: Issue in convergence proof in Appendix.D.
>
> **Thank you very much for check our proof carefully and find this inconsistent notation issue.**
>
> We recognize the algorithm in convergence proof has slight difference with Alg.1.
>
> [1/2] In code implementation, we also tried the update strategy as Eq.16 (Update $v^{l+1}$ by $u^l$ not $u^{l+1}$). It turns out that their **numerical accuracy is nearly identical**. The only minor disadvantage is, updating by Eq.16 has a slightly slower convergence than Alg.1. Thus we prove the convergence for this version, whose analysis is more tractable.
>
> [2/2] We are actively working on establishing a rigorous convergence proof for exactly Alg.1, which requires new techniques other than Hilbert Metric.
>
> Leveraging the Lyapunov Function, we make a proof sketch by analyzing convergence in dual space.
>
> The **potential function** of MLOT is $f(x,y)=\sum_k\sum_{ij} e^{-C_{ij}/\epsilon}\cdot e^{x_ki+y_kj}-\sum_{k}\langle x_k,a_k\rangle+\langle y_k,a_{k+1}\rangle$, where $\{x_k,y_k\}$ is dual var., and we denote $a_1=s,a_{K-1}=t$, and initialize with $x^1=y^1=\mathbf{0}$
>
> The **max possible declining volume** for potential func is $f(0,0)-f(x^\star,y^\star)\leq (K-1)\log(s/l)$, where $s$ is the max sum among $S_k$, and $l$ is the min entry among $S_k$.
>
> The **declining step of potential func** $f(x^{t-1},y^{t-1})-f(x^t,y^t)$ is $\sum_k |P_k^{t-1}|-|P_k^t| + KL(a_k^t||r(P_k^t)) + KL(a_{k+1}^t||c(P_k^t))$, where $r(\cdot),c(\cdot)$ means computing row/col sum. (This is similar process from [3] Lemma.2)
>
> We acknowledge the difficulty in analyzing the first term, the uncertainty of $a_k$ makes the L1-sum of $P_k$ keep changing, and cannot be eliminated. The key here is to bounded it to $\leq\Delta_k$, and $\Delta=\sum_k\Delta_k$.
>
> By Pinsker's inequality, we get the **declining step by one iteration** is **at least** $\frac{\epsilon'^2}{4(K-1)}-\Delta$.
>
> Thus Alg.1 **at most need** $(K-1)\log(s/l) / (\frac{\epsilon'^2}{4(K-1)}-\Delta)$ iterations **to get a $\epsilon'$ error solution**, indicates convergence.
>
> (Here we use L1-norm to capture convergence error: $\epsilon'=\sum_k |a_k-r(P_k)| + |a_{k+1}-c(P_k)|$)
>
> ---
> [3] Near-linear time approximation algorithms for optimal transport via Sinkhorn iteration.

---

### Official Review · Reviewer_PWyh · 2025-10-31

**Soundness:** 3
**Presentation:** 2
**Contribution:** 2
**Rating:** 4
**Confidence:** 3

**Summary:**

This paper proposes Multi-Layered Optimal Transport (MLOT), which extends standard Kantorovich optimal transport from a two-layer bipartite structure to a multi-layer framework.
The authors develop the MLOT-Sinkhorn algorithm that iteratively computes both couplings and intermediate distributions under entropic regularization.
In their empirical study, they demonstrate computational advantages over Gurobi and GraphOT-based methods on synthetic data and apply MLOT to data augmentation tasks including CLIP-based retrieval and visual graph matching.

Claim:I don't have much expertise in OT theory and entropic OT, hence I will focus more on the evaluation of applications of MLOT. I might also miss something when evaluating the comparison against existing literature in the field.

**Strengths:**

1. The formulation of MLOT seems novel to me.
2. They provide convergence analysis for their MLOT-Sinkhorn algorithm.
3. They demonstrate the efficiency and effectiveness of their method on synthetic datasets.
4. The writing of the theory part is overall clear and the paper is easy to follow.

**Weaknesses:**

1. I am somewhat confused about the setup of the original MLOT (Eq. 2). As far as I understand, the problem does not have a unique solution. Could the authors clarify this? Is this a misunderstanding? If not, would this lead to an ill-defined problem?
2. The application section is confusing. In Section 3.3, the authors start with "the application of MLOT to address tasks that involve augmented data." However, the authors didn't explain well what task it is and what role data augmentation plays in it. This leads to difficulty in understanding how MLOT could help and be a potentially better method.
3. Regarding the application of CLIP-based text-image retrieval, what is the intuition of setting text as the intermediate distribution? Why would solving for the unique solution to this MLOT-Sinkhorn problem lead to a better retrieval?
4. I am not convinced about the benefit of introducing the distribution of augmented data and the formulation of MLOT. Could the authors provide any insight on why this is fundamentally a better method? Regarding the empirical advantage of MLOT with Random Augmentation in Table 3, I feel like this is not a fair comparison since MLOT essentially gains more information from data. For example, for Softmax or Independent Sinkhorn, what if we compute scores using original data and augmented data separately, then aggregate them?

Overall, I find the theory part of this paper sound, but the application section and following experiments design are confusing.

**Questions:**

1. Could the authors elaborate on the relationship between MLOT and multi-marginal OT?
2. I am confused about Figure 4. I cannot tell why MLOT is better than Barycenter from the qualitative result.
3. I suggest moving Figure 2 to where it is first referred to.
4. The authors should use separate notation for their K and the metric R@k.

---

> ### Author Response · Authors · 2025-11-20
> **Response to reviewer (Part 1)**
>
> **We thank the reviewer for the thoughtful question and the opportunities to further explain some intuition and why MLOT is fundamentally better than compared baselines.** (Especially in response to W2 & W4)
>
> > W1: Original MLOT Eq.(2) does not have unique solution. Would this lead to an ill-defined problem?
>
> [1/3] Unregularized MLOT，**same as Classic OT, is LP**, which has no general unique solution guarantee indeed. But this **does not mean** MLOT or classic OT is ill-defined.
>
> [2/3] Same as Classic OT, for most cases **in practice**, the solution of MLOT actually lands on vertex of the feasible convex hull (not on edges, which means unique solution).
>
> [3/3] Theoretically, the uniqueness depends on property of cost metric $c(x,y)$. For example, Brenier Theorem shows the classic OT has unique solution under $||\mathcal{X}-\mathcal{Y}||^2$, and MLOT has similar property.
> (Plus, Entropic regularization makes the problem strictly convex, and the solution unique.)
>
>
> > W2: In the application of MLOT, what task and what role data augmentation plays in it?
>
> [1/5] In current paper, we show downstream application involved **(self)supervised representation learning**.
>
> [2/5] Data augmentation is widely adopted in contrastive learning (CL) by using **InfoNCE loss** (e.g. CLIP). InfoNCE formulates representation learning as a **softmax classification problem**, and pulls positive pairs together and pushes negative pairs apart.
>
> [3/5] The paper OT-CLIP[1] interprets CL through **point set matching**, rather than the above **pair-wise view**. Thus **CL can be formulated as a (inverse)OT problem** between two set. [1] further prove that **Softmax is the optimal solution of this special EntropicOT problem.**
>
> [4/5] Extending the line of OT-CLIP[1], MLOT is used as a **multiple layered Softmax** in our application. Leverage augmented data to formulate a chain-transport problem A-B-A', where A/B/A' is pos/neg point sets in CL problem.
>
> [5/5] In conclusion, what's new in the application part is that, we propose a framework that **contrast mulitiple pos/neg samples at same time**. To achieve this, we **formulate this as MLOT problem**. Vanilla Softmax cannot solve this, since it's bi-layer. **MLOT functions as a multi-layered Softmax** in our application.
>
> > W3: What is the intuition of setting text as the intermediate distribution? Why would solving for the unique solution to this MLOT-Sinkhorn problem lead to a better retrieval?
>
> [1/4] Given $k$ images, which $k$ of total $n_{captions}$ captions will be retrieved is unknown. Thus we **cannot formulate a classic OT problem, since the target distribution is unknown.**
>
> [2/4] The widely-adopted **practice** is to treat target distribution as **uniform**, and forcefully transport to it. This, to some degree, works but **lacks real significance.**
>
> [3/4] Our intuition comes from here, MLOT has unknown intermediate distribution that can be **adpatively computed**. Put the **two fixed known** distribution ($k$ images and its $k$ augmented candidates) as source/target, and **put the captions (to be retrieved) as intermediate**. This formulation has no extra approximate assumptions.
>
> [4/4] In Image->Text task, we set text as intermediate, while set image as intermediate in Text->Image task.
>
> > W4.1: What's the benefit of introducing the distribution of augmented data and the formulation of MLOT. Could the authors provide insight on why this is fundamentally a better method?
>
> [1/3] Introducing augmented data to formulate MLOT brings 2 benefits. One is change the problem from "transport from **given distrib** to **unknown distrib**" to "transport from **given distrib** to **given distrib**".
>
> [2/3] Another advantage is that, compared with two separate OT which **still needs assume intermediate as uniform**, MLOT hides the "to be retrieved distrib." into middle latent layer, which can be automatically optimized.
>
> [3/3] To summerize, by exploiting augmented data, **MLOT models the target distribution through latent intermediate layers**, so we can adaptively optimize it. In contrast, classical OT depends on fixed distribution, **whether introduced augmented data or not**, it has to make **wrong prior assumption on retrieved data.**
>
> ---
> [1] Understanding and Generalizing Contrastive Learning from the Inverse Optimal Transport Perspective. ICML.2023.

---

> > ### Author Response · Authors · 2025-11-20
> > **Response to reviewer (Part 2)**
> >
> > > W4.2: For Softmax or Independent Sinkhorn, what if we compute scores using original data and augmented data separately, then aggregate them?
> >
> > [1/3] Image/text augmentation is **static data processing**. And in retrieval task we only do this in **inference stage**.
> >
> > [2/3] MLOT maybe is suspected of profiting from extra augmented data. We add augmented data for Independent Sinkhorn, and also use $(P_1+P_2)/2$ to predict.
> >
> > [3/3] Results are as follow. It's true that leveraging extra information makes some progress, but still falls short of MLOT. This means MLOT makes fundamental progress beyond simply benefits from extra info.
> >
> > |          |             |          | Text→Image |          |          | Image→Text |          |
> > |:--------:|:----------- |:--------:|:----------:|:--------:|:--------:|:----------:|:--------:|
> > |          |             |   R@1    |    R@5     |   R@10   |   R@1    |    R@5     |   R@10   |
> > | ViT-B/32 | Indep. OT   |   32.1   |    58.0    |   68.5   |   46.4   |    71.5    |   79.8   |
> > |          | Separate OT |   33.3   |    58.9    |   70.2   |   50.5   |  **75.4**  | **83.3** |
> > |          | MLOT        | **35.1** |  **61.2**  | **72.2** | **50.7** |    75.1    | **83.3** |
> > | RN50x64  | Indep. OT   |   39.1   |    64.7    |   74.4   |   56.3   |    78.9    |   86.4   |
> > |          | Separate OT |   41.9   |    69.1    |   78.7   |   57.3   |    80.4    |   86.7   |
> > |          | MLOT        | **43.1** |  **70.3**  | **79.6** | **58.0** |  **81.1**  | **88.1** |
> >
> >
> > > Q1: Relationship between MLOT and Multi-Marginal OT?
> >
> > MLOT has very different objective and motivation from MMOT.
> >
> > [1/2]  MMOT seeks for a joint coupling, that can couple multiple **given** distributions in a **high-dimensional** space.
> >
> > [2/2] MLOT is extension from classic OT with multiple **steps or inner layers**. MLOT uses these unknown latent intermediate distributions to capture/adapt exsiting complex transport structure.
> >
> > > Q2: In Fig.4, why MLOT is better than Barycenter?
> >
> > [1/2] We think the second row is a **smoother** transition, especially col.4. Here "smooth" means the whole picture **transfers in a consistent level**, rather than some part of the pic transform faster while other part slower.
> >
> > [2/2] However, the more important thing we want to emphasize is the efficiency. MLOT **calculates the whole image path in a single run.** Each layer represents one middle image, and is ouput all at once. In contrast, barycenter based method needs vary weights to generate each interpolation.
> >
> > > Q3: Move Fig.2 to where it's first referred to.
> >
> > We take your suggestion seriously.
> >
> > We initially placed Fig.2 in early pages because it depicts the structure of the synthetic dataset, which may provides an intuitive overview of the MLOT formulation.
> >
> > > Q4: Confused Notation.
> >
> > Thanks very much for pointing out. We have fixed the confusion marks here.

---

> > > ### Comment · Reviewer_PWyh · 2025-11-27
> > >
> > > Thank you for the detailed response. My concerns have been addressed.
> > >
> > > I have adjusted my score accordingly, assuming that the authors will incorporate these explanations and intuitions (W2,W3,W4) into the final manuscript.

---

> ### Author Response · Authors · 2025-12-01
> **Thanks for response**
>
> We're glad to know your concerns have been addressed.
>
> We appreciate your instructive questions very much, which helps us further improve our expression.
>
> We have **completely re-written Section3.3** to better demonstrate "the motivation of introducing aumented data", "why MLOT is required" and "how MLOT is fundamentally better".

---

### Official Review · Reviewer_PCY1 · 2025-11-01

**Soundness:** 3
**Presentation:** 3
**Contribution:** 3
**Rating:** 8
**Confidence:** 3

**Summary:**

This work introduces multi-layer optimal transport (MLOT), extending classical entropic OT so that mass flows through $K-2$ intermediate layers that the marginal distributions are inferred jointly with the couplings between adjacent layers. Authors proposed two algoroithms to  compute the MLOT, a Bregman projection scheme and a Sinkhorn-like fixed-point iterations, which supports GPU-friendly matrix scaling while estimating the intermediate distributions. The authors provided convergence arguments and demonstrate the potential applications on synthetic layered transport, CLIP-based zero-shot retrieval with data augmentation, visual graph matching, and image interpolation. I reviewed this work from the previous conference cycle; the current draft shows clear progress with better exposition and richer experiments, so I inclined to support acceptance.

**Strengths:**

- The algorithm implementations are straightforward to implement on GPUs and appear to scale to reasonably large synthetic instances.

- The empirical comparison with generic LP solvers and shortest-path reductions highlights practical efficiency gains when $K>2$ and problem sizes exceed the comfort zone of exact solvers.

- This work showcased multiple applications of ML task including retrieval, graph matching, and interpolation indicate potential for applying MLOT.

**Weaknesses:**

- The formulation in Eq.\ (2) is a standard multi-stage min-cost flow with entropic regularization, which related formulations exist under the names layered OT, and network-flow OT. The paper does not clearly delineate how MLOT differs from these treatments beyond recomputing intermediate marginals.
-  The downstream sections would benefit from extra sentences on how CLIP features are normalized into histograms and how $(P_1,P_2)$ is converted into a retrieval score, plus a short comment on the fine-tuning schedule.
- I would appreciate a short sensitivity study on $K$ or $(\epsilon,\tau)$, and the reference list should include the GPU-friendly, graph-based OT solver of [1]. These adjustments seem feasible post-acceptance.

[1] Lahn et al.. “A Combinatorial Algorithm for Approximating the Optimal Transport in the Parallel and MPC Settings,” NeurIPS 2023.

**Questions:**

- Eq.~(8) features $(u_k\odot v_{k-1})^{-\epsilon/\tau}$ with little explanation can you elaborate more on this?
- How does MLOT fundamentally differ from multi-layer OT with entropic regularization beyond introducing unknown intermediate marginals?
-  For the CLIP retrieval task, how are continuous features mapped to probability vectors $a_k$ for each layer, and how is the final ranking extracted from $(P_1,P_2)$?
-  Could you report GPU memory details for the large synthetic problems as well?

---

> ### Author Response · Authors · 2025-11-20
>
> **We greatly appreciate your time and effort reviewing our paper.** Hope the following response addresses your concerns.
>
> > Q1: Eq.(8) with little explanation.
>
> The update of $a_k$ is obtained by differentiating the Lagrangian with respect to $a_k$, and enforcing the first-order optimality condition.
>
> > W1/Q2: Not clearly delineate how MLOT differs from layered OT and network-flow OT, beyond recomputing intermediate marginals.
>
> [1/3] Network-flow OT can be regarded as general Graph OT. Existing methods rely on computing shortest path on graph, either during the algorithm or before the algorithm.
>
> [2/3] MLOT aims to solve a special Graph OT problem, with layered structure (no cross-layer edges). MLOT-Sinkhorn **takes advantage of the prior layered structure**, without calculating shortest path.
>
> [3/3] We suppose the "Layered OT" refers to Multi-hierarchy Transport? In OR literature, there exists problem that models similar real-world scenario[2][3]. But they focus on **modeling real-world** as **accurate** as possible. As far as we know, no existing work has proposed a parallelizable algorithm for solving this.
>
> > Q3.1: How are continuous features mapped to probability vectors $a_k$ for each layer?
>
> [1/3] The distribution $\{a_k\}$ are not representing feature vector. In retrieval task, each value of $a_k$ represents one sample that to be retrieved. Each value in $s,t$ represents one given query.
>
> [2/3] Thus $s,t$ is set to be uniform distribution, which means each query is about to be matched once and only once. And $a_k$ has **unknow distribution**, because it **contains all candidates and not every captions are going to be retrieved.**
>
> [3/3] The feature vector is used for calculating similarity between sample pairs.
>
> > Q3.2: how $(P_1,P_2)$ is converted into a retrieval score / final ranking?
>
> [1/2] The matching confidence $P_1, P_2$ are regarded as retrieval score. And we use $P=(P_1+P_2^\top)/2$ for back-propagation since it's continuous.
>
> [2/3] In Graph Matching, the final ranking needs to be 0/1 matrix, and should not have **overlapping match**. We can realize this by applying **Hungarian algorithm** to get a unique 0/1 mapping from continuous transport coupling. **In practice**, however, we find that greedily choose argmax for each row works well, roughly satisfies the injection constraint.
>
> [3/3] In Retrieval, the evaluate metric is recall ratio. Thus we can directly choose top-k of each row in retrieval score $P$.
>
> > Q4: Could you report GPU memory details for the large synthetic problems?
>
> Of course. We use `torch.cuda.max_memory_allocated` to monitor **Peak Mem Usage**. $N$ means total points number, $K$ is layer number, and we set each layer's point number to be equal in the following test.
>
> | $N(\times10^3)$ |K=3 |K=5 |K=10 |
> |:-:| -:| -:| -:|
> |1|15.8MB|11.2MB|9.90MB|
> |4|132MB|58.1MB|36.7MB|
> |8| 499MB|209MB|119MB|
> |10|773MB|317MB|180MB|
> |20|2.99GB|1.20GB|698MB|
> |30|6.72GB|2.69GB|1.52GB|
> |40|11.9GB|4.78GB|2.69GB|
> |50|18.6GB|7.46GB|4.20GB|
> |60|>24GB|10.7GB|6.05GB|
>
> > W2: What's the fine-tuning schedule?
>
> [1/3] Take graph matching task as example，$P_1$ represents the matching score between A-B，$P_2$ represents the matching score between B-A$^\prime$. **Middle $a_2$ represents prediction of inlier distribution of B.**
>
> [2/3] We can introduce two new loss through MLOT. The first is consistency loss =KL($P_1, P_2^\top$). Because A' is augmentation of A, ideally, the key points matching solution should be identical.
>
> [3/3] The other one is **"inlier prediction loss"**. The motivation comes from [4]. [4] propose a pipline in deep-graph-matching inference stage: firstly predict total inliers, secondly use greedy method to match feature points. And [4] validates its effectiveness. MLOT can form inlier-prediction-loss by KL($a_2, a_2^{\text{ground-truth}}$), make this process able to back propagation.
>
> > W3: Reference include graph-based OT solver.
>
> As suggested by the reviewer, we add a comparison with the solver PyCoOT proposed in [1]. [1] proposed a parallel push-relabel method for solving OT. In the case of MLOT, it also relies on shortest path between $\mathbf{s,t}$. In the following table, shortest path (SP) is pre-computed by Dynamic Program (implemented in C++).
>
> We test on synthetic dataset with different layer number, $\mathcal{N}$ is set to 1K. MLOT has advantage in time efficiency, since no need for computing SP.
>
> |||MLOT($\tau=0$)|SP+CoOT|
> |-|-|-|-|
> |K=3|**Obj.**|0.6597|0.6561|
> ||**Time**|1.25|3.94|
> |K=5|**Obj.** |0.5486|0.5264|
> ||**Time**|1.35|1.70|
> |K=8|**Obj.**|1.2543|1.1612|
> ||**Time**|1.77|3.99|
> |K=10|**Obj.**|1.2162|1.1384|
> ||**Time**|1.83|2.64|
> |K=20|**Obj.**|3.6011|3.5508|
> ||**Time**|1.13|2.44|
> ---
> [2] A comprehensive survey of guaranteed-service models for multi-echelon inventory optimization
> [3] Simulation-Based Approach for Multi-Echelon Inventory System Selection:Case of Distribution Systems
> [4] Deep Learning of Partial Graph Matching via Differentiable Top-K.

---

### Author Response · Authors · 2025-11-30
**Summary of rebuttal and improvements, and thanks to AC**

Dear AC, thank you **very much** for taking the time to carefully read through the full review discussion. **We sincerely appreciate the effort and careful consideration you have dedicated to our submission (and other papers as well), especially under such urgent circumstances.**

If you have any questions or would like any additional clarification, we would be more than happy to provide further details.

**Summary of the rebuttal period and the reviewers’ updated positions:**
* Reviewer PCY1 (score:8, pending)

    + We **reply to all of reviewer's questions**. Particularly, **we supplement additional experiment** recording memory usage, and a graph-based OT baseline in varing K setting as reviewer asked.
    + We have **clarified the training schedule reviewer concerns and refine our expression in Section3.3.**

* Reviewer PWyh (score:4, raise to 6, concern addressed)

    The main concern lies in "why introduce augmented data, and why MLOT is fundamentally better?". We **comprehensively answered this question, added extra experiment, and re-write the Introduction part and Section3.3** as reviewer asked. The reviewer has **already adjusted score to 6.**

* Reviewer Fbf2 (score:4, pending)

    + The main concern lies in detailed relationship with Schrodinger Bridge(SB). We comprehensively **clarify the differences between MLOT, LowRankOT and SBP and add this part in Appendix.K.** Thus the 2D benchmark of SBP is not compatible with MLOT by this different formulation.

    + Reviewer points out the convergence proof inconsistent issue in Appendix.D, however, this inconsistency does not reflect a conceptual change of the algorithm, but only a **slightly different ordering** in the iteration. We also **implemented and tested the variant used in the proof, and found it achieved almost identical accuracy** to the original version.

    + We're actively working on the rigorous proof of exact Algorithm.1. We have briefly **outlined our promising progress in response** by drawing on techniques from the RAS algorithm and Lyapunov Function, and will continue this line of work.

* Reviewer gBDq (score:4, positive response, pending)

    There're several detailed questions, including theoretical properties, motivation of using augmented data, and intuition of MLOT works in supervised representation learning tasks. We **comprehensively responded to all of them**, and reviewer positively acknowledge **"most concerns have been addressed"** and point out **one last question** (disadvantage of aumented data). We also responded to this question and thus **addressed all of reviewer's concerns.**


**Summary of the improvements we have made to the manuscript:**
* Consider that 2 reviewers(PWyh,gBDq) concern the intuition of using augmented data and MLOT, and 1 reviewer(PCY1) concern our detailed training schedule, we **clarify the motivation in Introduction part**, and ***completely*** **re-write Section3.3 to more clearly explain why introducing augmented data and why MLOT is better.**

* As reviewer(gBDq) asked, we supplement the **proof of MLOT's convexity and overall time complexity** in Appendix.L & M

* As reviewer(Fbf2) asked, we clarify the detailed **difference between Schrodinger Bridge and MLOT** in Appendix.K

* **Fix several typo / expression**, as mentioned by 2 reviewers(PWyh,gBDq).

* **Refine our expression** in Experiment part.

Plus, we wish to state clearly that we have had **absolutely no contact with any reviewers outside the official reviewing system**. All behavior strictly followed the anonymous review protocol.

**Finally, we again deeply appreciate your work on our submission and remain fully available to address any further questions you may have.**

---

### Meta-Review · Area_Chair_tctA · 2026-01-07

**Summary:**

The paper introduces a new formulation of OT as a multi-stage transport problem along several intermediate locations, generalizing to some extents appraoches such as factored coupling. As such, most of the reviewers agree about the originality of the approach. Several concerns were raised regarding clarity, intuitions and details of convergence proofs of the proposed method. Authors answered those points in the rebuttal, partially addressing some of them (notably the convergence proof). Beyond those aspects, it is not fully clear in which general cases this formulation of OT is interesting. While the data augmentation part is of interest, a general discussion about the choice of the intermediate costs C_k and how they are designed seem to be critically missing from the paper. Given that a clear proof of convergence for Alg. 1 is still under examination, I am enclined to think that the paper is not ready for publication yet. I encourage authors in a subsequent submission to better detail how the intermediate costs are designed.

**Reviewer Concerns:**

Addressed concern: clarity, elements of motivations and relations to dynamic Ot formulation / Schrödinger bridges

Outstanding concern not resolved:
missing proof for the main algorithm.
"why MLOT is required" and "how MLOT is fundamentally better": still not clear

**Reviewer Scores:**

PWyh could have raised to 6

---

### Decision · Program_Chairs · 2026-01-26

Reject